# Beyond Test-Time Memory:
# State-Space Optimal Control for LLM Reasoning

**Peihao Wang** [1]  **Shan Yang** [2]  **Xijun Wang** [2]  **Tesi Xiao** [2]  **Xin Liu** [2]  **Changlong Yu** [2]  **Yu Lou** [2]
**Pan Li** [3]  **Zhangyang Wang** [1]  **Ming Lin** [4]  **René Vidal** [5]

⌂ vita-group.github.io/TTC-Net

## Abstract

Associative memory has long underpinned the design of sequential models. Beyond recall, humans reason by *projecting future states and selecting goal-directed actions*, a capability that modern language models increasingly require but do not natively encode. While prior work uses reinforcement learning or test-time training, planning remains external to the model architecture. We formulate reasoning as *optimal control* and introduce the *Test-Time Control (TTC)* layer, which performs finite-horizon *LQR* planning over latent states at inference time, represents a *value function* within neural architectures, and leverages it as the nested objective to enable *planning before prediction*. To ensure scalability, we derive a hardware-efficient LQR solver based on a symplectic formulation and implement it as a fused CUDA kernel, enabling parallel execution with minimal overhead. Integrated as an adapter into pretrained LLMs, TTC layers improve mathematical reasoning performance by up to **+27.8%** on MATH-500 and **2-3×** Pass@8 improvements on AMC and AIME, demonstrating that embedding optimal control as an architectural component provides an effective and scalable mechanism for reasoning beyond test-time training.

## 1. Introduction

Sequence-processing architectures have evolved from recurrent neural networks (RNNs) (Hochreiter & Schmidhuber,

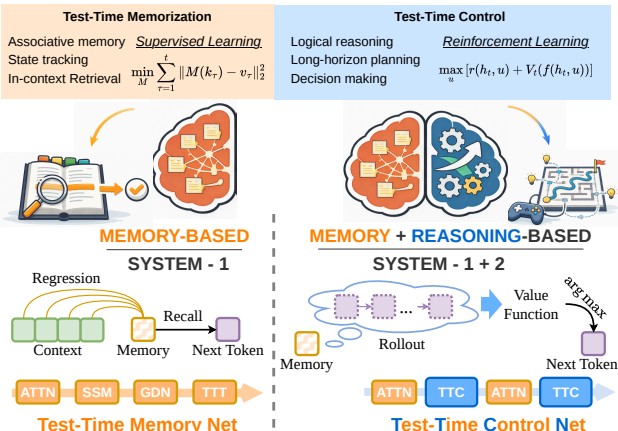

*Figure 1.* **Memory-only prediction vs. unified memory-control planning.** Memory-based models resemble human System 1 processing, relying on associative retrieval and producing fixed responses. Inspired by System 2 cognition, we model reasoning as an explicit optimal control problem and propose **TTC-Net**, a unified architecture that integrates test-time control (**TTC**) layers to encode value functions during sequential modeling, enabling planning before prediction.

1997; Sutskever et al., 2014; Cho et al., 2014) to transformers (Vaswani et al., 2017) and, more recently, to State-Space Models (SSMs) and linear RNNs (Gu & Dao, 2023; Yang et al., 2023b; 2024a; Beck et al., 2024). Despite substantial architectural differences, these models share a common design principle: prediction through *associative memory*. Past context is encoded as memory states, and the next token is generated by retrieving or decoding information from this stored representation (Hopfield, 1982; Hopfield & Tank, 1985; Widrich et al., 2020; Wang et al., 2025b). Attention mechanisms explicitly retain all past key-value pairs and match queries against them (Sun et al., 2024), while linear RNNs and SSMs progressively compress historical context into a fixed-size latent state via online regression and decode this state to produce future predictions (Schlag et al., 2021; Yang et al., 2024b; Behrouz et al., 2024; 2025a;c).

This memory-centric paradigm has proven highly effective for language modeling, yet increasingly reveals limitations when models are required to *reason*, *discover*, or *solve prob-*

[1] University of Texas at Austin [2] Amazon [3] Georgia Institute of Technology [4] University of Maryland at College Park [5] University of Pennsylvania. Correspondence to: Peihao Wang <peihaowang@utexas.edu>.

*Proceedings of the 43rd International Conference on Machine Learning*, Seoul, South Korea. PMLR 306, 2026. Copyright 2026 by the author(s).

*lems*. These tasks demand mechanisms beyond memorization and retrieval. From a cognitive perspective, human intelligence operates through an interplay between System 1 and System 2 thinking (Kahneman, 2011). System 1 relies on fast, automatic pattern matching over memory, while System 2 engages in deliberate, multi-step planning and long-horizon reasoning. Current LLM architectures largely instantiate System 1 behavior: they predict the next token by extrapolating from past context, but lack a dedicated architectural mechanism for System 2-style planning.

Recent advances in reinforcement learning (RL) have partially addressed this gap by making language models more goal-directed and planning-oriented, particularly in mathematical reasoning and coding (Shao et al., 2024; Guo et al., 2025). However, RL is typically applied as an external training or post-training procedure. The reward-driven optimization is absent during early pretraining and remains decoupled from the model's core inference mechanism (Hatamizadeh et al., 2025; Kobayashi et al., 2025). As a result, RL alone cannot fully overcome the reasoning ceilings imposed by memory-based architectures (Yue et al., 2025). The model learns *what* to optimize, but not *how* to reason through planning as part of its forward computation.

### 1.1. Main Contributions

Motivated by this gap in reasoning, we propose a different architectural perspective: ***viewing reasoning and planning as an optimal control problem over internal representations***. Rather than treating planning as a training-time or test-time optimization procedure, we internalize it directly into the model architecture. We operationalize this perspective architecturally by introducing a *Test-Time Control (TTC)* layer, which maps memory representations to optimal actions of a tractable control problem. Given a context-encoded latent state, a TTC layer performs test-time planning by solving a Markov Decision Process (MDP) with linear state transitions and quadratic cost functions, corresponding to a *Linear-Quadratic Regulator (LQR)*. The first-step optimal control action is then decoded as the next-token representation, enabling the model to *plan before prediction*.

Crucially, the TTC layer performs *online world modeling* (Ha & Schmidhuber, 2018) and endows each language modeling block with a latent *value function*. Environment dynamics and cost functions are synthesized from context, allowing planning to adapt to the current reasoning state. To capture temporal structure and long-horizon objectives, the TTC layer supports time-heterogeneous parameterizations of both dynamics and costs. Existing fast-weight test-time memorization methods (Yang et al., 2024b; Sun et al., 2024; Behrouz et al., 2025b) interpret inference as test-time estimation, solving self-supervised regression problems to better encode past context. In contrast, TTC interprets inference as test-time decision making, solving a structured optimal control problem to select actions that optimize future outcomes.

To enable end-to-end learning, we derive a differentiable formulation of the TTC layer inspired by differentiable optimization (Amos & Kolter, 2017). By casting the LQR problem into a KKT system, gradients can be propagated through the optimal solution by solving a second, structurally related LQR system (Amos et al., 2018). This results in a nested learning process (Behrouz et al., 2025b): an inner loop that solves the control problem conditioned on the current context, and an outer loop that updates the world-modeling parameters to improve downstream objectives.

A central challenge in integrating optimal control into large-scale language models is computational efficiency. Classical Riccati-based solvers (Bellman, 1954; Kalman et al., 1960) require sequential backward passes and repeated matrix inversions, making them poorly aligned with modern accelerators. We address this challenge through a hardware–algorithm co-design. Leveraging the symplectic structure of LQR dynamics (Bittanti et al., 1991; Laub, 2003), we reformulate the solver by replacing sequential linear-system solves with recursive matrix products and parallel matrix inversions, and further reduce the number of matrix inversions to a constant using structured parameterization. We further fuse this solver into a numerically stable CUDA kernel that efficiently supports both forward planning and backward differentiation.

Building on these components, we propose a hybrid architecture, *TTC-Net*, which interleaves TTC layers with memory-based modules such as attention. Moreover, TTC layer can be inserted as an adapter into pretrained LLMs without modifying the pre-trained base architecture. Empirically, TTC-Net consistently outperforms purely memory-based models on challenging reasoning tasks, including symbolic and mathematical problem solving. We demonstrate that TTC-Net achieves consistent gains on Sudoku games when trained from scratch and improves mathematical reasoning benchmarks up to +27.8% on MATH-500 and 2-3$\times$ Pass@8 improvements on AMC and AIME when inserted and fine-tuned as an adapter to base LLMs.

More broadly, this work proposes a unified view of training and reasoning in language models. Instead of treating supervised learning, RL, and test-time reasoning as separate stages, we internalize goal-directed reasoning as a hardware-efficient optimal control layer within the model architecture – integrating test-time memorization, world modeling, model-based RL, and planning in a single architectural framework, enabling models to reason over future trajectories at inference time while maintaining RL as a persistent objective throughout all training stages (Hatamizadeh et al., 2025; Dong et al., 2025).

## 2. Background: Memory-Based Architectures

Given a sequence of $t$ tokens $s_1, \cdots, s_t$, auto-regressive LLMs learn to predict the next token according to the conditional distribution $p_{\mathsf{LLM}}(s_{t+1} | s_1, \cdots, s_t)$. While a variety of architectures have been proposed to model this conditional probability, many of them are derived from associative memory mechanisms (Hopfield, 1982; 1984; Hopfield & Tank, 1985; Schmidhuber, 1992; Ramsauer et al., 2020). In this view, patterns associated with the context $s_1, \cdots, s_t$ are stored in memory, and $s_{t+1}$ is synthesized by retrieving information from the contextual representation through the minimization of an energy function.

To be more specific, let $\boldsymbol{X} = [\boldsymbol{x}_1, \cdots, \boldsymbol{x}_n]^\top \in \mathbb{R}^{n \times d}$ denote a sequence of token embeddings, and let $\boldsymbol{X}_{\leq t} = [\boldsymbol{x}_1, \cdots, \boldsymbol{x}_t]^\top$ denote the prefix up to time $t$. A memory unit is a sequence-to-sequence mapping, whose forward pass can be formulated as learning an online predictor $M_t : \mathbb{R}^d \to \mathbb{R}$ that encodes the patterns contained in $\boldsymbol{X}_{\leq t}$ and retrieves relevant information for a given query as the output. The memory is updated online by optimizing a self-supervised regression objective that stores predictive features from the observed prefix:

$$M_t = \arg\min_{M} \frac{1}{2} \sum_{\tau=1}^{t} w_{t,\tau} \|M(\boldsymbol{k}_\tau) - v_\tau\|_2^2 + R(M), \quad (1)$$

where $\boldsymbol{k}_\tau = \boldsymbol{W}_K \boldsymbol{x}_\tau$ and $\boldsymbol{v}_\tau = \boldsymbol{W}_V \boldsymbol{x}_\tau$ are two views created from the token $\boldsymbol{x}_\tau$ through projection matrices $\boldsymbol{W}_K \in \mathbb{R}^{d \times d}$ and $\boldsymbol{W}_V \in \mathbb{R}^{1 \times d}$, $w_{t,\tau} \in \mathbb{R}$ are coefficients re-weighing the past tokens, and $R(\cdot)$ is a regularizer over $M_t$. After obtaining the optimal predictor $M_t$, a query vector is formed as $\boldsymbol{q}_t = \boldsymbol{W}_Q \boldsymbol{x}_t$ with projection $\boldsymbol{W}_Q \in \mathbb{R}^{d \times d}$, and the memory unit produces $M_t(\boldsymbol{q}_t)$ as the output at the $t$-th position[1]. Because models built around this memory module perform an implicit regression over the historical context during the forward pass, they are often considered as *Test-Time Training (TTT)* architectures that can mitigate domain shift and prolong learning at inference time (Liu et al., 2024; Sun et al., 2024; Behrouz et al., 2024; Tandon et al., 2025). Different neural architectures can be interpreted as different instantiations of this paradigm:

**Attention.** As one of the most popular neural components, attention (Vaswani et al., 2017) has been recognized as a memory mechanism as well (Ramsauer et al., 2020). Under our formulation, attention can be viewed as a non-parametric regressor minimizing Eq. (1) with a mean squared error via the online Nadaraya-Watson estimator (Nadaraya, 1964; Sun et al., 2024). In particular, we define $w_{t,\tau} = 1$ for every $t, \tau$, and $R(M) \equiv 0$ in Eq. (1). Then attention can be

formulated via the following solution to Eq. (1):

$$M_t(\boldsymbol{q}) = \sum_{\tau=1}^{t} \frac{\exp(\boldsymbol{q}^\top \boldsymbol{k}_\tau / \sqrt{d}) v_\tau}{\sum_{\tau'=1}^{t} \exp(\boldsymbol{q}^\top \boldsymbol{k}_{\tau'} / \sqrt{d})}, \quad (2)$$

where scaled exponential function $\exp(\boldsymbol{q}^\top \boldsymbol{k} / \sqrt{d})$ is chosen as the kernel function. Grounded in non-parametric regression, attention has to traverse through all past tokens (KV caches) before making predictions for the next token.

**Linear RNNs.** Moving beyond attention, linear RNNs have been proposed as an alternative to avoid the notorious quadratic computational complexity of attention (Gu & Dao, 2023; Sun et al., 2023; Yang et al., 2023b; 2024a; Beck et al., 2024). In test-time training perspective, linear RNNs correspond to parametric solvers of Eq. (1), where $M_t$ becomes a linear mapping parameterized by $\boldsymbol{h}_t \in \mathbb{R}^d$, such that $M_t(\boldsymbol{q}) = \boldsymbol{h}_t^\top \boldsymbol{q}$. In linear RNNs, $\boldsymbol{h}_t$ is referred to as the *state*, which tracks all past tokens within a fixed-size memory block. By setting $w_{t,\tau} = 1$ only for $t = \tau$, leveraging gradient descent to update $\boldsymbol{h}_t$, and flexibly configuring $R(\cdot)$ for Eq. (1), a variety of linear RNNs or SSMs naturally emerge under the following general formulation (Gu & Dao, 2023; Sun et al., 2023; Yang et al., 2023b):

$$\boldsymbol{h}_t = \boldsymbol{A}_t \boldsymbol{h}_{t-1} + \boldsymbol{B}_t \boldsymbol{x}_t, \quad (3)$$

where $\boldsymbol{A}_t \in \mathbb{R}^{d \times d}$ controls what information will be passed to the next state, while $\boldsymbol{B}_t \in \mathbb{R}^{d \times d}$ encodes the raw inputs to the state space. Different linear RNNs are designed with distinct structures of $\boldsymbol{A}_t$ and $\boldsymbol{B}_t$. For instance, when $R(M)$ is removed from the objective Eq. (1) and gradient descent is applied with learning rate $\eta_t$, we obtain a new recurrent rule for $\boldsymbol{h}_t$ with $\boldsymbol{A}_t = (\boldsymbol{I} - \eta_t \boldsymbol{k}_t \boldsymbol{k}_t^\top)$, $\boldsymbol{B}_t = \eta_t \boldsymbol{k}_t \boldsymbol{W}_V^\top$. This corresponds to DeltaNet, as proposed in Schlag et al. (2021); Peng et al. (2025a); Yang et al. (2024b)[2].

## 3. A Planning-Based Neural Architecture

### 3.1. Learning to Reinforce Learning at Test Time

Our key idea is to model a layer for next-token prediction as the optimal decision induced by a tractable Markov Decision Process (MDP) environment, thereby programming a goal-seeking objective into the LLM architecture. Suppose we are predicting the $(\tau + 1)$-th token given the prior context embeddings $[\boldsymbol{x}_1, \cdots, \boldsymbol{x}_\tau]$, we begin by denoting an initial state as $\boldsymbol{h}_0 \in \mathbb{R}^d$, which encodes the context $[\boldsymbol{x}_1, \cdots, \boldsymbol{x}_\tau]$ up to time $\tau$. Given a planning horizon $T > 0$, we denote the latent state and token prediction at $t$ steps ahead as $\boldsymbol{h}_t, \boldsymbol{u}_t \in \mathbb{R}^d$, respectively. The goal is to output the first-step prediction that achieves optimality of the following

---

[1]Note that it is easy to extend $M_t$ to be a vector-to-vector mapping (e.g., vector-valued value entries in attention) by considering multiple $M_t$ functions solving Eq. (1) with shared key and query vectors.

[2]Similarly, by considering multiple parallel $M_t$'s, $\boldsymbol{h}_t$ can be extended to matrix-valued states as in Dao & Gu (2024); Yang et al. (2023b).

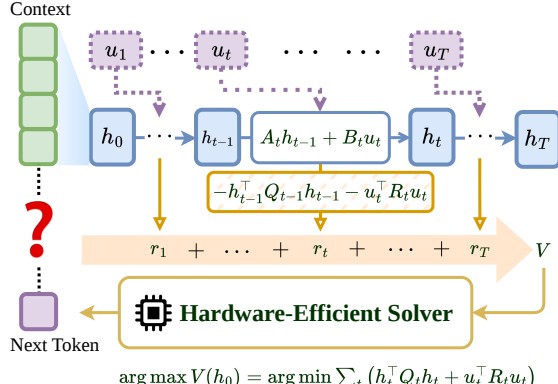

Context

$u_1 \cdots u_t \cdots \cdots u_T$

$h_0 \cdots h_{t-1} \boxed{A_t h_{t-1} + B_t u_t} \; h_t \cdots h_T$

$\boxed{-h_{t-1}^\top Q_{t-1} h_{t-1} - u_t^\top R_t u_t}$

$r_1 + \cdots + r_t + \cdots + r_T$    $V$

**? Hardware-Efficient Solver**

Next Token

$$\arg\max V(h_0) = \arg\min \sum_t \left( h_t^\top Q_t h_t + u_t^\top R_t u_t \right)$$

*Figure 2.* **Test-Time Control (TTC) layer.** To predict the next token, TTC layers think through a receding-horizon LQR problem with linear state evolution and a cost function. A *hardware-efficient solver* computes optimal actions using structured matrix operations, enabling test-time control with minimal inference overhead.

Bellman equation: $\boldsymbol{u}_1^* = \arg\max_{\boldsymbol{u}} Q_0(\boldsymbol{h}_0, \boldsymbol{u})$:

$$Q_t(\boldsymbol{h}_t, \boldsymbol{u}) = r_t(\boldsymbol{h}_t, \boldsymbol{u}) + V_{t+1}(f_{t+1}(\boldsymbol{h}_t, \boldsymbol{u})), \quad (4)$$

$$V_t(\boldsymbol{h}_t) = \max_{\boldsymbol{u}} Q_t(\boldsymbol{h}_t, \boldsymbol{u}), \quad (5)$$

where $0 \le t \le T - 1$, $f_t : (\boldsymbol{h}_{t-1}, \boldsymbol{u}_t) \mapsto \boldsymbol{h}_t$ models the state transition, $r_t(\boldsymbol{h}_t, \boldsymbol{u}_{t+1})$ is the reward function, $Q_t$ and $V_t$ are known as Q-function and value function, respectively. We let $V_T(\boldsymbol{h}_T) = r_T(\boldsymbol{h}_T)$ as the boundary condition. Solving a general Bellman equation as a neural network layer is computationally infeasible.

Instead, we can consider a more tractable yet expressive family of Eq. (4). Specifically, we adopt a linear dynamical system to model the state transition and optimize a quadratic reward function:

$$f_t(\boldsymbol{h}_{t-1}, \boldsymbol{u}_t) = \boldsymbol{A}_t \boldsymbol{h}_{t-1} + \boldsymbol{B}_t \boldsymbol{u}_t, \quad (6)$$

$$r_t(\boldsymbol{h}_t, \boldsymbol{u}_{t+1}) = -(\boldsymbol{h}_t^\top \boldsymbol{Q}_t \boldsymbol{h}_t + \boldsymbol{u}_{t+1}^\top \boldsymbol{R}_{t+1} \boldsymbol{u}_{t+1}), \quad (7)$$

where $\{(\boldsymbol{A}_t \in \mathbb{R}^{d \times d}, \boldsymbol{B}_t \in \mathbb{R}^{d \times d})\}_{1 \le t \le T}$ are matrices that govern the evolution of the state induced by the upcoming token $\boldsymbol{u}_t$, and $\{(\boldsymbol{Q}_t \in \mathbb{R}^{d \times d}, \boldsymbol{R}_t \in \mathbb{R}^{d \times d})\}_{1 \le t \le T}$ are symmetrical positive semi-definite matrices defining quadratic cost functions over the state $\boldsymbol{h}_t$ and action token $\boldsymbol{u}_t$. In particular, $\boldsymbol{Q}_0 = \boldsymbol{0}$ and $r_T(\boldsymbol{h}_T) = -\boldsymbol{h}_T^\top \boldsymbol{Q}_T \boldsymbol{h}_T$ for the boundary cases. Despite the linearity of the state transition function, the linear dynamical systems (Eq. (6)) have been shown to be sufficiently powerful to represent a broad family of MDPs (Hafner et al., 2019; Gu et al., 2020; 2021b;a). The objective in Eq. (8) naturally accommodates both process rewards $r_t(\boldsymbol{h}_t, \boldsymbol{u}_{t+1})$ through $\{(\boldsymbol{Q}_t, \boldsymbol{R}_t)\}_{1 \le t \le T-1}$ and an outcome reward $r_T(\boldsymbol{h}_T)$ via $\boldsymbol{Q}_T$ at the terminal step. Fig. 2 illustrates the design of our layer.

In fact, the MDP system in Eq. 4 is equivalent to solving a constrained optimization over the unrolled states and

forecast tokens for decisions $\boldsymbol{u}_1, \cdots, \boldsymbol{u}_T$:

$$\min_{\boldsymbol{u}_1, \cdots, \boldsymbol{u}_T} \frac{1}{2} \sum_{t=1}^{T} (\boldsymbol{h}_t^\top \boldsymbol{Q}_t \boldsymbol{h}_t + \boldsymbol{u}_t^\top \boldsymbol{R}_t \boldsymbol{u}_t) \quad (8)$$

$$s.t. \quad \boldsymbol{h}_t = \boldsymbol{A}_t \boldsymbol{h}_{t-1} + \boldsymbol{B}_t \boldsymbol{u}_t, \quad 1 \le t \le T.$$

Among all $\boldsymbol{u}_1^*, \cdots, \boldsymbol{u}_T^*$, we take the first-step optimal decision $\boldsymbol{u}_1^*$ as the output for the next token representation. Eq. (8) is the well-studied receding-horizon *Linear-Quadratic Regulator (LQR)* (Bellman, 1954; Kalman et al., 1960). The solution to an LQR problem in Eq. (8) is known as:

**Proposition 3.1** (Riccati Iteration). *The optimal $\boldsymbol{u}_1^*$ minimizing Eq. (8) depends linearly on $\boldsymbol{h}_0$ as $\boldsymbol{u}_1^* = \boldsymbol{K}_1^* \boldsymbol{h}_0$, where:*

$$\boldsymbol{K}_t^* = -(\boldsymbol{R}_t + \boldsymbol{B}_t^\top \boldsymbol{P}_t \boldsymbol{B}_t)^{-1} \boldsymbol{B}_t^\top \boldsymbol{P}_t \boldsymbol{A}_t, \quad (9)$$

$$\boldsymbol{P}_t = \boldsymbol{Q}_t + \boldsymbol{A}_{t+1}^\top \boldsymbol{P}_{t+1} \boldsymbol{A}_{t+1} - (\boldsymbol{B}_{t+1}^\top \boldsymbol{P}_{t+1} \boldsymbol{A}_{t+1})^\top$$
$$(\boldsymbol{R}_{t+1} + \boldsymbol{B}_{t+1}^\top \boldsymbol{P}_{t+1} \boldsymbol{B}_{t+1})^{-1} (\boldsymbol{B}_{t+1}^\top \boldsymbol{P}_{t+1} \boldsymbol{A}_{t+1}), \quad (10)$$

*for $t = 1, \cdots, T$, and $\boldsymbol{P}_T = \boldsymbol{Q}_T$.*

In addition, we can show that value function $V_t(\boldsymbol{h}_t) = -\frac{1}{2} \boldsymbol{h}_t^\top \boldsymbol{P}_t \boldsymbol{h}_t$ (see Appendix A.1). Thus, $\boldsymbol{P}_t$ is often called a value matrix. Eq. (10) computing the value matrices $\{\boldsymbol{P}_t\}_{1 \le t \le T}$ is also known as the Riccati iteration (Bellman, 1954; Kalman et al., 1960). It involves a backward iteration from time $T$ to 1 to compute the value of $\boldsymbol{P}_1$ and $\boldsymbol{K}_1$. This implies that we are not able to solve $\boldsymbol{u}_1^*$ by only considering past or local information at time $t = 1$ because $\boldsymbol{u}_1^*$ has long-term influence over all subsequent states.

Since the forward pass of our proposed module decodes a memory state $\boldsymbol{h}_0$ into the optimal decision $\boldsymbol{u}_1^*$ by internally solving an optimal control problem, we refer to our proposed module as a *Test-Time Control (TTC)* layer. TTC layer is parameterized via $\{(\boldsymbol{A}_t, \boldsymbol{B}_t, \boldsymbol{Q}_t, \boldsymbol{R}_t)\}_{t=1}^T$, and we denote it end-to-end as $\texttt{TTC}\left(\boldsymbol{h}_0, \{(\boldsymbol{A}_t, \boldsymbol{B}_t, \boldsymbol{Q}_t, \boldsymbol{R}_t)\}_{t=1}^T\right)$. Unlike a memory-based layer that minimizes regression objectives over historical data points, a TTC layer optimizes for future trajectories and minimizes costs incurred in the long shot. Solving for the optimal action $\boldsymbol{u}_t^*$ during the model forward requires computing the value function on the fly at inference time. This process endows each sequential modeling block with an internal value function $V_t$ and a nested model-based RL objective.

### 3.2. Differentiating TTC Layers

To train a TTC layer end-to-end, suppose we obtain the output $\boldsymbol{o} = \texttt{TTC}(\boldsymbol{h}_0, \{(\boldsymbol{A}_t, \boldsymbol{B}_t, \boldsymbol{Q}_t, \boldsymbol{R}_t)\}_{t=1}^T)$ and compute loss $\ell(\boldsymbol{o})$ in forward pass. Then, during the backward pass, we receive gradients $\nabla_{\boldsymbol{o}} \ell$ backpropagated from the overall loss function. Our goal is to leverage this gradient to compute gradients with respect to the TTC parameters $\{(\boldsymbol{A}_t, \boldsymbol{B}_t, \boldsymbol{Q}_t, \boldsymbol{R}_t)\}_{t=1}^T$.

To show how this can be achieved, we need to demonstrate that the TTC objective (Eq. (8)) can be cast into a KKT form: $\mathcal{L} = \frac{1}{2}\sum_{t=1}^{T}(\boldsymbol{h}_t^\top \boldsymbol{Q}_t \boldsymbol{h}_t + \boldsymbol{u}_t^\top \boldsymbol{R}_t \boldsymbol{u}_t) + \sum_{t=1}^{T}\boldsymbol{\lambda}_t^\top (\boldsymbol{A}_t \boldsymbol{h}_{t-1} + \boldsymbol{B}_t \boldsymbol{u}_t - \boldsymbol{h}_t)$ where $\boldsymbol{\lambda}_t \in \mathbb{R}^d$ are defined as Lagrangian multipliers, also referred to as *co-states* in LQR. A standard approach to solving the KKT system for $\{(\boldsymbol{h}_t^*, \boldsymbol{u}_t^*, \boldsymbol{\lambda}_t^*)\}_{t=1}^T$ is to first compute the optimal control $\boldsymbol{u}_1^*$ via Riccati iteration (Proposition 3.1), and then alternatively compute the new state $\boldsymbol{h}_t$ via Eq. (6) and the next optimal control via $\boldsymbol{u}_{t+1} = \boldsymbol{K}_{t+1}\boldsymbol{h}_t$ where $\boldsymbol{K}_{t+1}$ is as defined in Eq. (9). Afterwards, another backward scanning is performed to obtain co-states through this relation: $\boldsymbol{\lambda}_t = \boldsymbol{A}_{t+1}^\top \boldsymbol{\lambda}_{t+1} + \boldsymbol{Q}_t \boldsymbol{h}_t$, which is derived from the KKT condition $\frac{\partial \mathcal{L}}{\partial \boldsymbol{h}_t} = \boldsymbol{0}$. See Appendix A.2 for the derivation.

With these notions, we can represent gradients w.r.t. $\{(\boldsymbol{A}_t, \boldsymbol{B}_t, \boldsymbol{Q}_t, \boldsymbol{R}_t)\}_{t=1}^T$ and $\boldsymbol{h}_0$ using states, actions, and co-states via the approach introduced in Amos & Kolter (2017); Amos et al. (2018).

**Theorem 3.2.** *Consider a TTC layer with parameters* $\{(\boldsymbol{A}_t, \boldsymbol{B}_t, \boldsymbol{Q}_t, \boldsymbol{R}_t)\}_{t=1}^T$ *and initial state $\boldsymbol{h}_0$, suppose we have output $\boldsymbol{o} = \text{TTC}(\boldsymbol{h}_0, \{(\boldsymbol{A}_t, \boldsymbol{B}_t, \boldsymbol{Q}_t, \boldsymbol{R}_t)\}_{t=1}^T)$, the loss is computed as $\ell(\boldsymbol{o})$, and we have obtained $\nabla_{\boldsymbol{o}}\ell$. Then the gradients w.r.t. $\{(\boldsymbol{A}_t, \boldsymbol{B}_t, \boldsymbol{Q}_t, \boldsymbol{R}_t)\}$ are:*

$$\nabla_{\boldsymbol{A}_t}\ell = \boldsymbol{\lambda}_t^* \widetilde{\boldsymbol{h}}_{t-1}^{*\top} + \widetilde{\boldsymbol{\lambda}}_t^* \boldsymbol{h}_{t-1}^{*\top}, \quad \nabla_{\boldsymbol{B}_t}\ell = \boldsymbol{\lambda}_t^* \widetilde{\boldsymbol{u}}_t^{*\top} + \widetilde{\boldsymbol{\lambda}}_t^* \boldsymbol{u}_t^{*\top},$$

$$\nabla_{\boldsymbol{Q}_t}\ell = \frac{1}{2}(\widetilde{\boldsymbol{h}}_t^* \boldsymbol{h}_t^{*\top} + \boldsymbol{h}_t^* \widetilde{\boldsymbol{h}}_t^{*\top}), \nabla_{\boldsymbol{R}_t}\ell = \frac{1}{2}(\widetilde{\boldsymbol{u}}_t^* \boldsymbol{u}_t^{*\top} + \boldsymbol{u}_t^* \widetilde{\boldsymbol{u}}_t^{*\top}),$$

*and gradient w.r.t. $\boldsymbol{h}_0$ can be computed by $\nabla_{\boldsymbol{h}_0}\ell = \widetilde{\boldsymbol{\lambda}}_0^*$, where $\{(\boldsymbol{h}_t^*, \boldsymbol{\lambda}_t^*, \boldsymbol{u}_t^*)\}_{t=1}^T$ are the solution to the KKT system associated with Eq. (8) and $\{(\widetilde{\boldsymbol{h}}_t^*, \widetilde{\boldsymbol{\lambda}}_t^*, \widetilde{\boldsymbol{u}}_t^*)\}_{t=1}^T$ are the solution to the KKT system corresponding to the following LQR system with $\widetilde{\boldsymbol{h}}_0 = \boldsymbol{0}$:*

$$\min \frac{1}{2}\sum_{t=1}^{T}(\widetilde{\boldsymbol{h}}_t^\top \boldsymbol{Q}_t \widetilde{\boldsymbol{h}}_t + \widetilde{\boldsymbol{u}}_t^\top \boldsymbol{R}_t \widetilde{\boldsymbol{u}}_t) + \nabla_{\boldsymbol{o}}\ell^\top \widetilde{\boldsymbol{u}}_1 \quad (11)$$

$$s.t. \quad \widetilde{\boldsymbol{h}}_t = \boldsymbol{A}_t \widetilde{\boldsymbol{h}}_{t-1} + \boldsymbol{B}_t \widetilde{\boldsymbol{u}}_t, \quad 1 \le t \le T.$$

We refer to the LQR solved in the forward pass as the *primal LQR*, and the additional LQR solved during the backward pass as the *dual LQR*. Compared to the primal LQR, the dual LQR assumes zero initial state while introducing an additional affine term $\nabla_{\boldsymbol{o}}\ell^\top \widetilde{\boldsymbol{u}}_1$ in the cost function. Despite these differences, solving the primal and dual LQRs shares the same underlying algorithm (see Appendix A for a unified derivation to account for the affine term). The gradients with respect to the primal LQR parameters are then obtained by combining the KKT solutions from both the primal and dual LQRs, using Kronecker product.

### 3.3. Hardware Co-Design for TTC

As seen in Sec. 3.1 and 3.2, one inference iteration of TTC requires solving one LQR while one training iteration re-quires solving two LQRs. In practice, incorporating TTC layers into LLMs requires both training and inference to operate over thousands of tokens and long planning horizons. This setting makes conventional solvers based on Riccati iteration (Proposition 3.1) (Amos et al., 2018; Hansen et al., 2023) difficult to apply. We observe that Riccati iteration is a simultaneously compute- and I/O-bound algorithm. (1) Riccati iteration computes $\boldsymbol{P}_t$ sequentially, requiring a matrix inversion at each time step. This leads to a computational complexity of $O(Td^3)$. Such inversions are poorly supported by dedicated hardware accelerators and cannot be parallelized across the horizon due to inherent sequential dependencies. (2) Each Riccati iteration also involves a sequence of matrix multiplications. These operations incur substantial memory traffic between GPU high-bandwidth memory (HBM) and on-chip SRAM, resulting in significant I/O overhead. To address these challenges, we propose a new solver that improves parallelizability and hardware utility for solving LQR problems.

**Hardware-Efficient LQR Solver.** Our new algorithm solving LQR is shown by the following result:

**Theorem 3.3** (Symplectic Iteration)**.** *The first-step solution to* $\min \sum_{t=1}^{T} \left[ \frac{1}{2}(\boldsymbol{h}_t^\top \boldsymbol{Q}_t \boldsymbol{h}_t + \boldsymbol{u}_t^\top \boldsymbol{R}_t \boldsymbol{u}_t) + \boldsymbol{r}_t^\top \boldsymbol{u}_t \right]$ *s.t.* $\boldsymbol{h}_t = \boldsymbol{A}_t \boldsymbol{h}_{t-1} + \boldsymbol{B}_t \boldsymbol{u}_t$ *can be computed by* $\boldsymbol{u}_1^* = -\boldsymbol{R}_1^{-1}(\boldsymbol{B}_1^\top (\boldsymbol{A}_1^\top)^{-1}\boldsymbol{Y}_1^{-1}\boldsymbol{Y}_2(\boldsymbol{h}_0 + \boldsymbol{y}_3) + \boldsymbol{r}_1)$ *such that:*

$$\begin{bmatrix} \boldsymbol{Y}_1 & \boldsymbol{Y}_2 \end{bmatrix} = \begin{bmatrix} \boldsymbol{I} & \boldsymbol{Q}_T \end{bmatrix} \prod_{t=1}^{T} \boldsymbol{\Sigma}_{T-t+1}, \quad (12)$$

$$\boldsymbol{y}_3 = \begin{bmatrix} \boldsymbol{I} & \boldsymbol{Q}_T \end{bmatrix} \sum_{t=1}^{T} \left( \prod_{r=1}^{T-t} \boldsymbol{\Sigma}_{T-r+1} \right) \begin{bmatrix} \boldsymbol{0} \\ -\boldsymbol{B}_t \boldsymbol{R}_t^{-1} \boldsymbol{r}_t \end{bmatrix},$$

$$\boldsymbol{\Sigma}_t = \begin{bmatrix} (\boldsymbol{A}_t^\top)^{-1} & (\boldsymbol{A}_t^\top)^{-1}\boldsymbol{Q}_{t-1} \\ \boldsymbol{G}_t(\boldsymbol{A}_t^\top)^{-1} & \boldsymbol{A}_t + \boldsymbol{G}_t(\boldsymbol{A}_t^\top)^{-1}\boldsymbol{Q}_{t-1} \end{bmatrix}, \quad (13)$$

$$\boldsymbol{G}_t = \boldsymbol{B}_t \boldsymbol{R}_t^{-1} \boldsymbol{B}_t^\top. \quad (14)$$

Theorem 3.3 yields a solver for both the primal and dual LQRs. Note that $\boldsymbol{\Sigma}_t$ is a symplectic matrix, and computing the first-step action $\boldsymbol{u}_1^*$ requires a reverse iterative matrix product from $t = T$ to $t = 1$ (see Eq. (12)). Such an iterative matrix product is called *(reverse) symplectic iteration*. A key observation from Theorem 3.3 is that it replaces the sequential Riccati recursion dominated by dense matrix inversions with a cumulative matrix product over $\boldsymbol{\Sigma}_t$, utilizing the symplectic structure of $\boldsymbol{\Sigma}_t$. Importantly, the matrix inverses appearing within each $\boldsymbol{\Sigma}_t$ are independent across time steps and can therefore be computed fully in parallel. Outside of this cumulative product, only a single dense matrix inversion is required to recover the optimal first-step decision $\boldsymbol{u}_1^*$. By Theorem 3.3, all remaining sequential computation is confined to matrix multiplication, which enables effective utilization of dedicated hardware

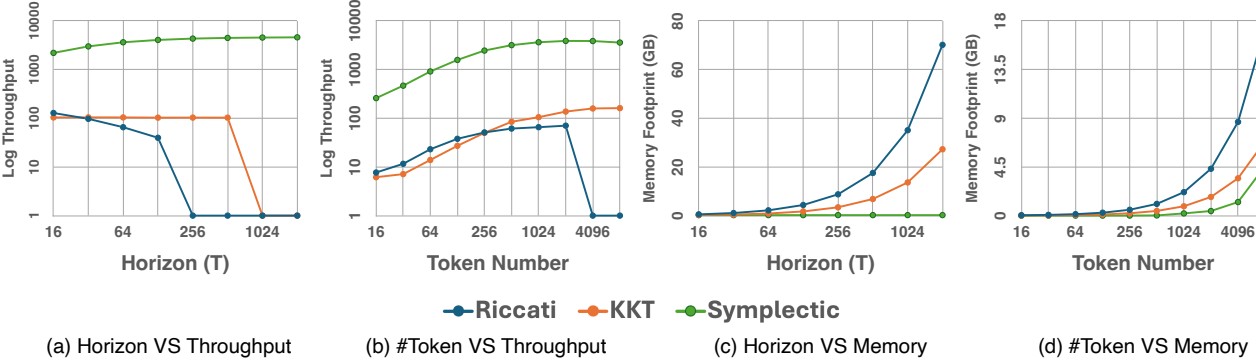

(a) Horizon VS Throughput  (b) #Token VS Throughput  (c) Horizon VS Memory  (d) #Token VS Memory

*Figure 3.* **Benchmarking running speed and memory of different LQR solvers**. Throughput is reported in TFLOPs/s on a logarithmic scale. Among all evaluated solvers, our method achieves over a **10x** higher throughput while maintaining constant memory cost w.r.t. horizon. Zero throughput indicates an out-of-memory error during execution.

accelerators (e.g., Tensor Cores) for high-throughput dense linear algebra. The algorithm implied from Theorem 3.3 to compute $\boldsymbol{u}_1^*$ is listed in Algorithm 3.

As seen in Theorem 3.2, backpropagation through TTC layers requires access to the full trajectories of both the primal and dual LQRs. As we will show in Theorem A.4, Appendix A, there exists a *forward symplectic iteration* involving the symplectic adjoint of $\boldsymbol{\Sigma}_t$ in Eq. (12) that computes each $(\boldsymbol{h}_t, \boldsymbol{u}_t, \boldsymbol{\lambda}_t)$ via an iterative matrix-vector product from $t = 1$ to $t = T$. This result provides a unified implementation for computing $\boldsymbol{u}_1^*$ and performing rollouts.

**Structured Parameterization.** In fact, we can further specialize to a structured family of LQRs in which $\{\boldsymbol{A}_t\}_{t=1}^T$ and $\{\boldsymbol{R}_t\}_{t=1}^T$ are defined to be diagonal matrices. This structure makes all inversion operations on $\boldsymbol{A}_t$ and $\boldsymbol{R}_t$ significantly more efficient, reducing the number of nontrivial dense matrix inversions from $O(T)$ to $O(1)$. We will adopt this efficient parameterization in our implementation of the TTC layer. As demonstrated in prior work (Gupta et al., 2022; Gu et al., 2022; Gu & Dao, 2023), diagonalized SSMs can be as expressive as dense SSMs. In contrast, diagonalization of $\boldsymbol{A}_t$ cannot simplify the computation for Riccati iteration as $\boldsymbol{P}_t$ remains a dense matrix in Eq. (10).

**Kernel Optimization.** The primary computation in the symplectic iteration algorithm (Theorem 3.3), which relies on sequential matrix multiplications, is well suited for kernel fusion. Accordingly, we fuse Eq. 12 into a CUDA kernel to reduce memory I/O traffic and further improve computational efficiency. We also optimize numerical stability and the backward pass. See details in Appendix B.

**Empirical Validation.** We benchmark the training throughput and memory footprint of our fused symplectic iteration solver and compare it against two classic baselines: (1) a native Riccati iteration solver with gradients computed via automatic differentiation; and (2) a KKT-based solver

(Amos et al., 2018) replacing gradient computation with a manner similar to Theorem 3.2. As shown in Fig. 3, the symplectic iteration solver achieves over a $10\times$ higher throughput than both the Riccati- and KKT-based solvers. Meanwhile, the two classic solvers are bottlenecked by memory overhead, whereas our solver scales much more efficiently with the planning horizon and the number of tokens to be processed.

## 4. TTC-Net: A Hybrid Model with TTC Layer

In this section, we introduce the key techniques that turn TTC layers into a language model.

**Contextualization.** Learning a single set of LQR parameters $\{(\boldsymbol{A}_t, \boldsymbol{B}_t, \boldsymbol{Q}_t, \boldsymbol{R}_t)\}_{t=1}^T$ shared across all test samples enforces a fixed decision pattern, which limits adaptability to specific contexts and disables flexible adjustment of the planning horizon. Meanwhile, constraining $\{(\boldsymbol{A}_t, \boldsymbol{B}_t, \boldsymbol{Q}_t, \boldsymbol{R}_t)\}_{t=1}^T$ to be uniform across the horizon further restricts expressivity, hindering the model's ability to capture temporal variations in the environment dynamics and cost functions (including discounting effects) (Nhu et al., 2025). Combining these observations, we propose to condition the TTC parameters on both the initial state $\boldsymbol{h}_0$, which encodes the context, and the time step $t$. Given an initial state $h_0$, we first generate a group of time modulation coefficients $\boldsymbol{\Gamma}_\square = \exp(-\operatorname{diag}(s_{\Gamma_\square}(\boldsymbol{h}_0)))$ where $\square \in \{A, B, Q\}$. Then we synthesize LQR parameters by modulating power series of $\boldsymbol{\Gamma}_\square$:

$$\boldsymbol{A}_t = \boldsymbol{I} + \boldsymbol{\Gamma}_A^t \operatorname{diag}(s_A(\boldsymbol{h}_0)), \quad \boldsymbol{B}_t = \sum_{i=1}^r s_B(\boldsymbol{h}_0)_i \boldsymbol{B}^{(i)} \boldsymbol{\Gamma}_B^t,$$

$$\boldsymbol{Q}_t = \boldsymbol{\Gamma}_Q^t \left( \sum_{i=1}^r s_Q(\boldsymbol{h}_0)_i \boldsymbol{Q}^{(i)} \right) \boldsymbol{\Gamma}_Q^t, \boldsymbol{R}_t = \operatorname{diag}(s_R(\boldsymbol{h}_0)),$$

where we single out the terminal cost parameter $\boldsymbol{Q}_T$ and override it with $\boldsymbol{Q}_T = \sum_{i=1}^r s_{Q_f}(\boldsymbol{h}_0)_i \boldsymbol{Q}^{(i)}$. $s_\square$ denotes a linear layer with task-specific activations for every $\square \in$

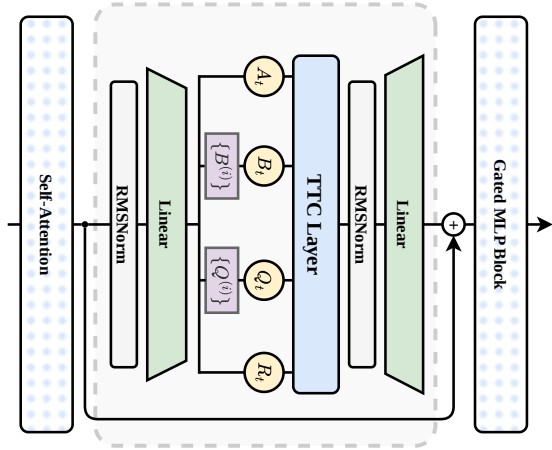

*Figure 4.* **Overview of TTC-Net.** We construct a hybrid model by inserting a TTC layer between attention and MLP.

$\{A, B, Q, R, Q_f, \Gamma_A, \Gamma_B, \Gamma_Q\}$. In particular, $s_{\Gamma_\square}$, $s_R$, and $s_Q$ employ a softplus activation, necessary for $\boldsymbol{Q}_t$ and $\boldsymbol{R}_t$'s positive semi-definiteness. $\boldsymbol{A}_t$ is parameterized with an added identity term to ensure invertibility in $\boldsymbol{\Sigma}_t$. $\{\boldsymbol{Q}^{(i)}\}_{i=1}^r$ and $\{\boldsymbol{B}^{(i)}\}_{i=1}^r$ form two groups of basis matrices, which are linearly combined using context-informed coefficients. Every $\boldsymbol{Q}^{(i)} = \boldsymbol{Q}_c^{(i)}\boldsymbol{Q}_c^{(i)\top}/\sqrt{d}$ adopts a Cholesky reparameterization with unconstrained $\{\boldsymbol{Q}_c^{(i)}\}_{i=1}^r$. All $\boldsymbol{\Gamma}_\square$ take values in $(0, 1)$, thereby discounting the effect of the state unrolled over the horizon. In practice, time-dependent parameters are not instantiated in HBM; instead they are computed on the fly within the kernel.

**Hybrid with Memory Layers.** The TTC layer operates by taking an expressive memory state as input. Consequently, TTC layers need to be interleaved with memory-based modules and form a hybrid architecture for language modeling. In practice, we insert a TTC layer between attention and MLP every 8 transformer blocks. We refer to the resulting architecture as *TTC-Net*. In TTC-Net, a TTC layer is applied independently to each token. TTC-Net employs a linear mapping to project context-rich token features produced by attention into the initial state of the TTC layer. The output of the TTC layer is then normalized and passed through another linear mapping $\boldsymbol{W}_{out}$, after which it is added back to the residual stream of the LLM backbone. Please refer to Fig. 4 for a quick overview and Appendix C for the formulation. TTC-Net can be trained either from scratch or by continuing training from a pretrained model. When fine-tuning from a pretrained model, we initialize the output projection $\boldsymbol{W}_{out} = \boldsymbol{0}$ so that the initial model remains identical to the original backbone.

**More Architecture Designs.** In addition to the core ingredients above, we defer our design of multi-head structures, parameterizations of contextualization modules, training strategies, and test-time scaling techniques to Appendix C.

*Table 1.* Accuracy (%) on Sudoku reasoning. Best method (ours) is marked in **bold** while the runner-up is underlined.

| Methods | Single-Step | | Multi-Step | |
|---|---|---|---|---|
| | Board Acc. | Cell Acc. | Board Acc. | Cell Acc. |
| Transformer | 58.50 | 86.54 | 90.10 | 94.08 |
| Mamba | 54.60 | 85.50 | 88.60 | 91.29 |
| Mamba2 | 55.50 | 85.10 | 87.20 | 90.52 |
| GDN | 57.30 | 87.19 | 89.80 | 93.70 |
| Samba | 57.20 | 87.99 | 90.40 | 94.61 |
| TTC-Net (Ours) | **61.30** | **90.17** | **93.40** | **97.33** |

## 5. Experiments

### 5.1. Sudoku Solving

Sudoku is a classical logical puzzle that requires long-horizon reasoning, constraint propagation, and multi-step planning to reach a valid solution, making it a canonical benchmark for evaluating structured reasoning and planning capabilities of deep learning models (Palm et al., 2018; Du et al., 2019; Wang et al., 2025a; Long, 2023). We leverage Sudoku as a controlled synthetic task on long-horizon reasoning to evaluate the effectiveness of TTC-Net.

**Data.** We adopt the 10k $9 \times 9$ boards from a challenging Sudoku dataset introduced by Palm et al. (2018), where each board contains between 17 and 34 given digits. We reserve 1k boards for evaluation and use the remaining 9k boards for training. Each Sudoku board is tokenized as a sequence, with each cell represented by a token from the vocabulary $\{[\texttt{mask}], 1, \ldots, 9\}$. These cell tokens are mapped to token embeddings, and the resulting sequence is passed into a sequential processing backbone.

**Baselines.** We compare TTC-Net against a range of popular neural architectures for sequential modeling: (1) a Transformer with pure attention blocks (Vaswani et al., 2017), (2) Mamba and Mamba2 (Gu & Dao, 2023; Dao & Gu, 2024), (3) GDN (Yang et al., 2024a), and (4) Samba (Ren et al., 2024). We adopt positional embeddings and full self-attention for transformer blocks. We include GDN as it has been shown to excel at state tracking (Grazzi et al., 2024; Peng et al., 2025a). Samba represents a hybrid architecture that combines the Mamba layer with sliding-window attention. All SSM, linear attention, and sliding-window attention layers scan the sequence twice bidirectionally to propagate information across the Sudoku grid. Despite their architectural differences, all of these baselines fall under the category of memory-based approaches. All models consist of 32 layers in total. A classifier is attached to the backbone to decode the representations into digit predictions.

**Training and Evaluation.** During training, we apply a cross-entropy loss over all masked cells. In addition, we decode the output of each block through the classifier and ap-

*Table 2.* **Accuracy (%) on math reasoning datasets** compared with baseline methods, including test-time memorization approaches. We mark the best performer in **bold** and the runner-up with underline. Our TTC-Net achieves the highest level of accuracy consistently.

| Models | MATH-500 | AMC | | AIME24 | | AIME25 | |
|---|---|---|---|---|---|---|---|
| | | Acc@8 | Pass@8 | Acc@8 | Pass@8 | Acc@8 | Pass@8 |
| Base model | 25.00 | 6.63 | 31.32 | 0.00 | 0.00 | 0.00 | 0.00 |
| Full Finetuning | 46.80 | 20.78 | 46.98 | 1.67 | 6.67 | 0.00 | 0.00 |
| + MLP (Houlsby et al., 2019) | 40.80 | 18.67 | 33.73 | 0.00 | 0.00 | 0.00 | 0.00 |
| + Attention (Vaswani et al., 2017) | 47.00 | 20.48 | 44.58 | 0.42 | 3.33 | 1.25 | 6.67 |
| + RetNet (Sun et al., 2023) | 42.60 | 19.58 | 39.76 | 2.50 | 13.33 | 0.00 | 0.00 |
| + Mamba (Gu & Dao, 2023) | 44.80 | 22.29 | 44.58 | 0.83 | 3.33 | 1.67 | 3.33 |
| + GDN (Yang et al., 2024a) | 47.80 | 17.77 | 37.35 | 0.42 | 3.33 | 0.83 | 6.67 |
| + MesaNet (von Oswald et al., 2025) | 47.40 | 12.65 | 27.71 | 1.25 | 10.00 | 0.00 | 0.00 |
| TTC-Net | **52.80** | **23.34** | **54.22** | **3.33** | **20.00** | **5.00** | **20.00** |

ply the same loss to supervise intermediate representations, encouraging the model to make progressive corrections at every layer, following Yang et al. (2023c). We list all training configurations in Appendix E.7. We evaluate the models using two approaches: (1) performing a single forward pass and measuring accuracy on the masked cells; and (2) iteratively completing one cell at a time by selecting the most confident prediction at each step, repeating this process until all cells are filled, akin to CoT (Wei et al., 2022) (see Algorithm 6). For both approaches, we report both cell- and board-level accuracy. Throughout training and testing, we fix $T_{train} = T_{test} = 4$ for TTC-Net.

**Results.** Tab. 1 reports the performance of all evaluated architectures. TTC-Net outperforms all baselines by a clear margin in accuracy. In particular, TTC-Net surpasses the strongest runner-up baseline, i.e., Transformer, by 2.8%, in board-level accuracy on single-step completion. Moreover, TTC-Net demonstrates highly coherent multi-step reasoning, as evidenced by its superior accuracy on multi-step Sudoku solving. In fact, this performance gain is expected, as TTC-Net is explicitly designed to internalize a long-horizon reasoning mechanism that is well-suited for solving Sudoku. Sudoku can be viewed as a constraint satisfaction problem (Wang et al., 2019; Yang et al., 2023c). The TTC objective (Eq. (8)) provides an effective approximation by casting Sudoku solving as a sequential decision-making process, where placing a digit on the board corresponds to a linear state transition, and the quadratic cost function serves as a smooth surrogate for constraint satisfaction barriers.

**5.2. Math Reasoning**

**Settings.** In this section, we evaluate the effectiveness of TTC-Net on mathematical reasoning tasks for LLM. Rather than training models from scratch, we adopt a continual learning setting in which we perform supervised fine-tuning (SFT) on pretrained models augmented with additional architectural modules. From this perspective, SFT can be

viewed as a form of imitation learning, and training the TTC layers corresponds to an instance of inverse RL, where TTC layers learn to infer latent state transition and objectives that facilitate cloning the expert behavior. We use `Llama-3-Instruct-7B` as the base model and initialize TTC-Net from the base model with zero initialization applied to the output projection (see Sec. 4). To enable a fair comparison, we construct hybrid architectures by inserting an additional memory layer with a comparable number of parameters between the attention and MLP every 8 transformer blocks. All newly introduced layers are trained jointly with the rest of the model. The new layers play a role similar to adapter modules that allocate additional learning capacity for acquiring new reasoning abilities. The memory mechanisms considered in this study include attention (Vaswani et al., 2017), RetNet (Sun et al., 2023), Mamba (Gu & Dao, 2023), GDN (Yang et al., 2024a), and MesaNet (von Oswald et al., 2025).

**Benchmark Evaluation.** We fine-tune all baselines for one epoch on the `OpenThoughts2-114K` dataset (Guha et al., 2025), along with an additional 800K self-collected and curated reasoning examples. See Appendix E.7 for more information about the curated dataset and the training hyperparameters. To disentangle improvements due to architectural modifications from those arising solely from fine-tuning, we additionally report the performance of supervised fine-tuning (SFT) applied only to the base model weights, without introducing new modules. We evaluate reasoning performance on a suite of challenging benchmarks, including Math-500 (Hendrycks et al., 2021), AMC (LI et al., 2024), AIME 2024, and AIME 2025. We adopt $T_{test} = 8, 16, 16$ at inference time for Math-500, AMC, and AIME, respectively. For AMC and AIME, we sample 8 responses per prompt using temperature sampling and report both Avg@8 and Pass@8 accuracy following Zuo et al. (2025). The results are summarized in Tab. 2. Across all benchmarks, TTC-Net consistently outperforms

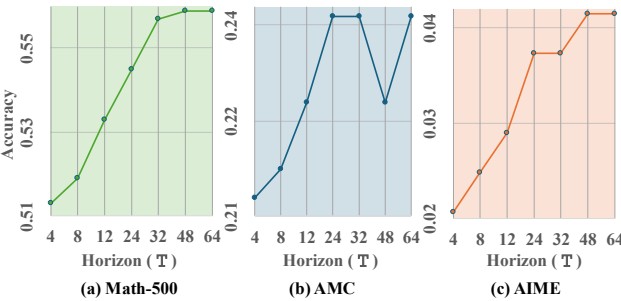

*Figure 5.* Test-time scaling with TTC layers. TTC allows scaling test-time compute to improve performance by enlarging the planning horizon $T$.

all other fine-tuned hybrid architectures. Notably, the base model achieves zero accuracy on the challenging AIME 2024 and AIME 2025 datasets, whereas TTC-Net exhibits clear performance emergence, highlighting its ability to unlock complex reasoning capabilities. In contrast, hybrid models augmented with additional memory layers fail to stably surpass SFT applied to the base model alone. Notably, TTC-Net achieves large gains in Pass@8 accuracy, suggesting that the TTC layer extends the effective reasoning boundary of the base model, in contrast to prior observations that post-training alone cannot overcome the intrinsic capability ceiling of the backbone (Yue et al., 2025). These results indicate that the control-based objective underlying the TTC layer provides a critical inductive mechanism for complex reasoning in LLMs.

**Test-Time Scaling.** As discussed in Sec. 4, the TTC layer supports a flexible planning horizon and can achieve improved performance with longer horizons. Accordingly, we evaluate the fine-tuned TTC-Net by extending its planning horizon at test time to examine how performance scales with $T$. The results, shown in Fig. 5, validate our hypothesis: reasoning accuracy consistently improves as the planning horizon increases. Notably, although the maximum horizon used during training is 32, the model generalizes successfully to $T = 64$ at test time and continues to enhance reasoning performance.

### 5.3. Ablation Studies

We conduct ablation studies to evaluate the main design choices of TTC-Net. The detailed results are deferred to Appendix E.1, while we summarize the findings below. Time-heterogeneous parameterization is important for both accuracy and horizon extrapolation: the time-homogeneous variant achieves $48.40\%$ at $T_{\text{test}} = 8$ and drops to $45.70\%$ at $T_{\text{test}} = 16$, while the full time-heterogeneous model improves to $52.80\%$ and $53.60\%$, respectively. Horizon sampling is also critical: fixed-horizon training performs reasonably at the matched horizon but degrades sharply from $50.60\%$ to $31.50\%$ when evaluated at a longer hori-

zon, whereas sampled horizons preserve extrapolation. The interleaving pattern affects the accuracy-cost trade-off. Distributing TTC layers more evenly with attention layers yields stronger performance than sparse or consecutive insertion. These results support our default design using time-heterogeneous control parameters, Poisson log-normal horizon sampling, and an $8{:}1$ attention-to-TTC ratio.

### 5.4. Additional Evaluations

We conduct several additional experiments: we compare TTC-Net against decoding-based test-time scaling methods (e.g., BoN and ToT), evaluate the fine-tuned models beyond math reasoning tasks, test its effectiveness on a stronger Qwen2.5-Math-7B backbone, analyze inference efficiency under different planning horizons, and analyze TTC latent trajectories. These experiments collectively examine the generality, efficiency, and interpretability of TTC-Net. Please find the full results and detailed analyses in Appendix E.

## 6. Conclusion

We introduce Test-Time Control (TTC)-Net, a new architectural paradigm that augments memory-based language models with test-time control. At its core, TTC-Net embeds TTC layers that formulate reasoning as an optimal control problem over internal representations, enabling models to plan future trajectories before decoding the next token. To make this paradigm scalable, we derive a hardware-efficient LQR solver that supports parallel execution with minimal overhead, allowing TTC layers to be deployed as lightweight adapters within pretrained LLMs. Empirically, TTC-Net consistently improves reasoning performance. More broadly, TTC-Net reframes test-time learning as structured decision making rather than memorization or parameter adaptation, unifying memory, world modeling, reinforcement learning objectives, and long-horizon planning within a single architecture.

**Limitations and Future Works.** Theoretically, although a single TTC layer has a well-defined and interpretable optimization objective, it remains unclear how multiple TTC layers jointly interact and represent dynamics within a deep transformer. A rigorous theoretical analysis from this perspective remains an open direction for future research. Empirically, it is worthwhile to explore more expressive parameterizations of the latent-space dynamics and reward modeling, including advanced linear approaches (Yang et al., 2024b) and even non-linear formulations (Amos et al., 2018), while maintaining compatibility with hardware-level optimization constraints. While our results demonstrate the promise of TTC layers as core components of LLMs, a more comprehensive evaluation across larger-scale models and all stages of training remains open for future work.

## Acknowledgements

PW thanks Liangzu Peng, Junbo Li, and Zun Wang for insightful discussions and for pointing to useful resources. PW also thanks Ruisi Cai for proofreading this manuscript. This work was done when PW was at Amazon. PW is in part supported by Google PhD Fellowship in Machine Learning and ML Foundations.

## Impact Statement

This paper presents work whose goal is to advance the field of Machine Learning. There are many potential scientific implications with societal benefits by this work, including reduced computation (thereby energy consumption) and lower memory requirement due to our new TTC layer and differentiable formulation, in addition to added capability of reasoning in language models.

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

# A. Theory

**Definition A.1** (Linear-Quadratic Regulator ([Kalman et al., 1960](#))). Given LQR system with parameters $\{(A_t, B_t, Q_t, R_t, r_t)\}_{t=1}^T$ and initial state $h_0 = h_{init}$, the optimization problem can be formally written as:

$$\min_{\substack{h_0, h_1, \cdots, h_T \\ u_1, \cdots, u_T}} \left[ \frac{1}{2}(h_t^\top Q_t h_t + u_t^\top R_t u_t) + r_t^\top u_t \right] \tag{15}$$

$$\text{s.t.} \quad h_{t+1} = A_{t+1} h_t + B_{t+1} u_{t+1}, \quad h_0 = h_{init} \tag{16}$$

Note that we distinguish between the notation for $h_0$ as an optimization variable and its assigned value $h_{init}$. In the main text, we use $h_0$ in place of $h_{init}$ without separate notations when no confusion arises.

Below, we introduce two approaches to solve Eq. ([15](#)), along the way, proving Proposition [3.1](#) and Theorem [3.3](#). Afterwards, we conclude this section by proving Theorem [3.2](#).

## A.1. Riccati Iteration

One of the most famous algorithms to solve Eq. ([15](#)) is through Riccati iteration algorithm, grounded by the following proposition:

**Proposition A.2.** *The optimal $u_t$ minimizing Eq. ([8](#)) depends affinely on $h_{t-1}$ as $u_t = K_t h_{t-1} + k_t$, where $K_t$ and $k_t$ are given by:*

$$K_t = -(R_t + B_t^\top P_t B_t)^{-1} B_t^\top P_t A_t, \qquad k_t = -(R_t + B_t^\top P_t B_t)^{-1}(B_t^\top p_t + r_t), \tag{17}$$

*where:*

$$P_t = Q_t + A_{t+1}^\top P_{t+1} A_{t+1} - (B_{t+1}^\top P_{t+1} A_{t+1})^\top (R_{t+1} + B_t^\top P_{t+1} B_t)^{-1}(B_{t+1}^\top P_{t+1} A_{t+1}), \tag{18}$$

$$p_t = A_{t+1}^\top p_{t+1} - A_{t+1}^\top P_{t+1} B_{t+1}(R_{t+1} + B_{t+1}^\top P_{t+1} B_{t+1})^{-1}(B_{t+1}^\top p_{t+1} + r_{t+1}), \tag{19}$$

*for $t = 1, \cdots, T$ with boundary condition $P_T = Q_T$ and $p_T = 0$.*

*Proof.* First, define the value function $V_t(h)$ as the minimal cost-to-go given $h_t = h$ for any $t \geq 0$:

$$V_t(h) = \min_{u_{t+1}, \cdots, u_T} \frac{1}{2} \left[ h_t^\top Q_t h_t + \sum_{s=1}^{T-t}(h_{t+s}^\top Q_{t+s} h_{t+s} + u_{t+s}^\top R_{t+s} u_{t+s} + 2r_{t+s}^\top u_{t+s}) \right]. \tag{20}$$

Unrolling the optimization problem in Eq. ([20](#)) by one step yields the Bellman equation:

$$V_t(h) = \min_u \left[ \frac{1}{2} h^\top Q_t h + \frac{1}{2} u^\top R_{t+1} u + r_{t+1}^\top u + V_{t+1}(A_{t+1} h + B_{t+1} u) \right].$$

Next, we make the inductive hypothesis that $V_{t+1}(h) = \frac{1}{2} h^\top P_{t+1} h + p_{t+1}^\top h + c_{t+1}$ for some symmetric matrix $P_{t+1} \in \mathbb{R}^{d \times d}$, vector $p_{t+1} \in \mathbb{R}^d$, and scalar $c_{t+1} \in \mathbb{R}$. This hypothesis holds for $t = T$: $V_T(h) = h^\top Q_T h$. Substituting the hypothesis into the Bellman equation gives

$$V_t(h) = \min_u \left[ \frac{1}{2} h^\top Q_t h + \frac{1}{2} u^\top R_{t+1} u + r_{t+1}^\top h + \frac{1}{2}(A_{t+1} h + B_{t+1} u)^\top P_{t+1}(A_{t+1} h + B_{t+1} u) \right.$$

$$\left. + p_{t+1}^\top(A_{t+1} h + B_{t+1} u) + c_{t+1} \right]$$

$$= \min_u \frac{1}{2} \left[ h^\top(Q_t + A_{t+1}^\top P_{t+1} A_{t+1})h + u^\top(R_{t+1} + B_{t+1}^\top P_{t+1} B_{t+1})u \right.$$

$$\left. + 2h^\top A_{t+1}^\top P_{t+1} B_{t+1} u + 2(B_{t+1}^\top p_{t+1} + r_{t+1})^\top u + 2p_{t+1} A_{t+1} h + 2c_{t+1} \right].$$

The minimizer admits the closed form

$$u_{t+1} = -(R_{t+1} + B_{t+1}^\top P_{t+1} B_{t+1})^{-1}(B_{t+1}^\top P_{t+1} A_{t+1} h + B_{t+1}^\top p_{t+1} + r_{t+1}), \tag{21}$$

and the minimum value is

$$V_t(\boldsymbol{h}) = \frac{1}{2}\boldsymbol{h}^\top \left[ \boldsymbol{Q}_t + \boldsymbol{A}_{t+1}^\top \boldsymbol{P}_{t+1} \boldsymbol{A}_{t+1} - (\boldsymbol{B}_{t+1}^\top \boldsymbol{P}_{t+1} \boldsymbol{A}_{t+1})^\top (\boldsymbol{R}_{t+1} + \boldsymbol{B}_t^\top \boldsymbol{P}_{t+1} \boldsymbol{B}_t)^{-1} (\boldsymbol{B}_{t+1}^\top \boldsymbol{P}_{t+1} \boldsymbol{A}_{t+1}) \right] \boldsymbol{h}$$

$$+ \left[ \boldsymbol{A}_{t+1}^\top \boldsymbol{p}_{t+1} - \boldsymbol{A}_{t+1}^\top \boldsymbol{P}_{t+1} \boldsymbol{B}_{t+1} (\boldsymbol{R}_{t+1} + \boldsymbol{B}_{t+1}^\top \boldsymbol{P}_{t+1} \boldsymbol{B}_{t+1})^{-1} (\boldsymbol{B}_{t+1}^\top \boldsymbol{p}_{t+1} + \boldsymbol{r}_{t+1}) \right]^\top \boldsymbol{h}$$

$$+ c_{t+1} - \frac{1}{2}(\boldsymbol{B}_{t+1}^\top \boldsymbol{p}_{t+1} + \boldsymbol{r}_{t+1})^\top (\boldsymbol{R}_{t+1} + \boldsymbol{B}_{t+1}^\top \boldsymbol{P}_{t+1} \boldsymbol{B}_{t+1})^{-1} (\boldsymbol{B}_{t+1}^\top \boldsymbol{p}_{t+1} + \boldsymbol{r}_{t+1}).$$

This completes the induction by identifying $\boldsymbol{P}_t, \boldsymbol{p}_t, c_t$ for $V_t$ as desired. □

*Remark* A.3. Proposition 3.1 is a special case of Theorem A.2 when considering $t = 1$ and no affine cost on actions. Solving an LQR with a linear cost upon actions is needed for backward (Sec. 3.2).

Theorem A.2 implies Algorithm 1 to compute all $\{(\boldsymbol{h}_t, \boldsymbol{u}_t)\}_{t=1}^T$.

---

**Algorithm 1** Finite-horizon LQR via Riccati recursion

---

**Require:** $\{(\boldsymbol{A}_t, \boldsymbol{B}_t, \boldsymbol{Q}_t, \boldsymbol{R}_t, \boldsymbol{r}_t)\}_{t=1}^T$, initial state $\boldsymbol{h}_{init}$
**Ensure:** actions $\{\boldsymbol{u}_t\}_{t=1}^T$, states $\{\boldsymbol{h}_t\}_{t=1}^T$
1: Initialize terminal parameters $\boldsymbol{P}_T \leftarrow \boldsymbol{Q}_T, \boldsymbol{p}_T \leftarrow \boldsymbol{0}$
2: **for** $t = T - 1, \ldots, 1$ **do**                                    ▷ reverse Riccati recursion
3:    $\boldsymbol{P}_t \leftarrow \boldsymbol{Q}_t + \boldsymbol{A}_{t+1}^\top \boldsymbol{P}_{t+1} \boldsymbol{A}_{t+1} - (\boldsymbol{B}_{t+1}^\top \boldsymbol{P}_{t+1} \boldsymbol{A}_{t+1})^\top (\boldsymbol{R}_{t+1} + \boldsymbol{B}_t^\top \boldsymbol{P}_{t+1} \boldsymbol{B}_t)^{-1} (\boldsymbol{B}_{t+1}^\top \boldsymbol{P}_{t+1} \boldsymbol{A}_{t+1})$
4:    $\boldsymbol{p}_t \leftarrow \boldsymbol{A}_{t+1}^\top \boldsymbol{p}_{t+1} - \boldsymbol{A}_{t+1}^\top \boldsymbol{P}_{t+1} \boldsymbol{B}_{t+1} (\boldsymbol{R}_{t+1} + \boldsymbol{B}_{t+1}^\top \boldsymbol{P}_{t+1} \boldsymbol{B}_{t+1})^{-1} (\boldsymbol{B}_{t+1}^\top \boldsymbol{p}_{t+1} + \boldsymbol{r}_{t+1})$
5: **end for**
6: Assign initial state $\boldsymbol{h}_0 \leftarrow \boldsymbol{h}_{init}$.
7: **for** $t = 1, 2, \ldots, T$ **do**                                    ▷ forward rollout
8:    $\boldsymbol{u}_t \leftarrow -(\boldsymbol{R}_t + \boldsymbol{B}_t^\top \boldsymbol{P}_t \boldsymbol{B}_t)^{-1} (\boldsymbol{B}_t^\top \boldsymbol{P}_t \boldsymbol{A}_t \boldsymbol{h}_{t-1} + \boldsymbol{B}_t^\top \boldsymbol{p}_t + \boldsymbol{r}_t)$
9:    $\boldsymbol{h}_t \leftarrow \boldsymbol{A}_t \boldsymbol{h}_{t-1} + \boldsymbol{B}_t \boldsymbol{u}_t$
10: **end for**
11: **return** $\{\boldsymbol{u}_t\}_{t=1}^T, \{\boldsymbol{h}_t\}_{t=1}^T$.

---

## A.2. KKT Reformulation

We introduce Lagrangian multiplier $\{\boldsymbol{\lambda}_t\}_{0 \le t \le T}$, and the objective Eq. (15) can be transformed to:

$$L = \frac{1}{2}\sum_{t=1}^T \left( \boldsymbol{h}_t^\top \boldsymbol{Q}_t \boldsymbol{h}_t + \boldsymbol{u}_t^\top \boldsymbol{R}_t \boldsymbol{u}_t \right) + \sum_{t=1}^T \boldsymbol{r}_t^\top \boldsymbol{u}_t + \sum_{t=0}^{T-1} \boldsymbol{\lambda}_{t+1}^\top (\boldsymbol{A}_{t+1} \boldsymbol{h}_t + \boldsymbol{B}_{t+1} \boldsymbol{u}_{t+1} - \boldsymbol{h}_{t+1}) + \boldsymbol{\lambda}_0^\top (\boldsymbol{h}_{init} - \boldsymbol{h}_0), \quad (22)$$

wherer $\boldsymbol{\lambda}_t$ is also called co-state. The dual problem to Eq. (15) can then be written as (Boyd & Vandenberghe, 2004):

$$\max_{\boldsymbol{\lambda}_0, \cdots, \boldsymbol{\lambda}_T} \inf_{\{(\boldsymbol{h}_t, \boldsymbol{u}_t)\}_{t=1}^T} L(\{(\boldsymbol{h}_t, \boldsymbol{\lambda}_t, \boldsymbol{u}_t)\}_{t=1}^T). \quad (23)$$

By KKT conditions (Boyd & Vandenberghe, 2004), the optimal solution $\{(\boldsymbol{h}_t, \boldsymbol{\lambda}_t, \boldsymbol{u}_t)\}_{t=1}^T$ to Eq. (23) must satisfy:

$$\frac{\partial L}{\partial \boldsymbol{h}_t} = \boldsymbol{Q}_t \boldsymbol{h}_t + \boldsymbol{A}_{t+1}^\top \boldsymbol{\lambda}_{t+1} - \boldsymbol{\lambda}_t = \boldsymbol{0}, \qquad 1 \le t \le T - 1, \quad (24)$$

$$\frac{\partial L}{\partial \boldsymbol{h}_t} = \boldsymbol{A}_{t+1}^\top \boldsymbol{\lambda}_{t+1} - \boldsymbol{\lambda}_t = \boldsymbol{0}, \qquad t = 0, \quad (25)$$

$$\frac{\partial L}{\partial \boldsymbol{h}_t} = \boldsymbol{Q}_t \boldsymbol{h}_t - \boldsymbol{\lambda}_t = \boldsymbol{0}, \qquad t = T \quad (26)$$

$$\frac{\partial L}{\partial \boldsymbol{u}_t} = \boldsymbol{R}_t \boldsymbol{u}_t + \boldsymbol{B}_t^\top \boldsymbol{\lambda}_t + \boldsymbol{r}_t = \boldsymbol{0}, \qquad 1 \le t \le T, \quad (27)$$

$$\frac{\partial L}{\partial \boldsymbol{\lambda}_t} = \boldsymbol{A}_t \boldsymbol{h}_{t-1} + \boldsymbol{B}_t \boldsymbol{u}_t - \boldsymbol{h}_t = \boldsymbol{0}, \qquad 1 \le t \le T, \quad (28)$$

$$\frac{\partial L}{\partial \boldsymbol{\lambda}_t} = \boldsymbol{h}_{init} - \boldsymbol{h}_t = \boldsymbol{0}, \qquad t = 0. \quad (29)$$

Combining Riccati iteration, Eq. (24), (25) and (26) imply a way to compute all $\{\lambda_t\}_{t=0}^T$ from $\{(h_t, u_t)\}_{t=1}^T$. See Algorithm 2 for more details.

---

**Algorithm 2** Extension to Riccati recursion

---

**Require:** $\{(A_t, B_t, Q_t, R_t, r_t)\}_{t=1}^T$, actions $\{u_t\}_{t=1}^T$, states $\{h_t\}_{t=0}^T$,
**Ensure:** co-states $\{\lambda_t\}_{t=0}^T$                                           $\triangleright$ Eq. (26)
  1: Initialize terminal co-state $\lambda_T \leftarrow Q_T h_T$
  2: **for** $t = T - 1, \ldots, 1$ **do**
  3:     $\lambda_t \leftarrow Q_t h_t + A_{t+1}^\top \lambda_{t+1}$                  $\triangleright$ Eq. (24)
  4: **end for**
  5: $\lambda_0 \leftarrow A_1^\top \lambda_1$                                       $\triangleright$ Eq. (25)
  6: **return** $\{\lambda_t\}_{t=0}^T$.

---

### A.3. Symplectic Iteration

We prove the symplectic iteration theorem in two parts. First, we establish the joint relation between states and co-states, which is useful for forward computation of all states and co-states starting from $h_0$ and $\lambda_0$.

**Theorem A.4** (Forward symplectic iteration). *Given LQR system in Eq. (15) with parameters $\{(A_t, B_t, Q_t, R_t, r_t)\}_{t=1}^T$ and initial state $h_{init}$. The following recursive relation between state $h_t$ and co-state $\lambda_t$ is always true for $1 \le t \le T$:*

$$\begin{bmatrix} h_t \\ \lambda_t \end{bmatrix} = S_t \begin{bmatrix} h_{t-1} \\ \lambda_{t-1} \end{bmatrix} + b_t, \qquad S_t = \begin{bmatrix} A_t + G_t(A_t^\top)^{-1}Q_{t-1} & -G_t(A_t^\top)^{-1} \\ -(A_t^\top)^{-1}Q_{t-1} & (A_t^\top)^{-1} \end{bmatrix}, \qquad b_t = \begin{bmatrix} -B_t R_t^{-1} r_t \\ 0 \end{bmatrix}, \qquad (30)$$

*where we define $Q_0 = 0$ and $G_t = B_t R_t^{-1} B_t^\top$.*

*Proof.* Starting with KKT condition Eq. (27), we have $u_t = -R_t^{-1}(B_t^\top \lambda_t + r_t)$. After re-organizing Eq. (28) and (24):

$$h_t = A_t h_{t-1} - B_t R_t^{-1} B_t^\top \lambda_t - B_t R_t^{-1} r_t, \qquad\qquad 1 \le t \le T, \qquad (31)$$
$$\lambda_t = Q_t h_t + A_{t+1}^\top \lambda_{t+1} \qquad\qquad 1 \le t \le T - 1, \qquad (32)$$

with boundary condition:

$$\lambda_0 = A_1^\top \lambda_1, \qquad\qquad \lambda_T = Q_T h_T, \qquad\qquad h_0 = h_{init}. \qquad (33)$$

By Eq. (32) above, we have:

$$\lambda_t = (A_t^\top)^{-1}(\lambda_{t-1} - Q_{t-1} h_{t-1}),$$

and henceforth

$$h_t = A_t h_{t-1} - B_t R_t^{-1} B_t^\top (A_t^\top)^{-1}(\lambda_{t-1} - Q_{t-1} h_{t-1}) - B_t R_t^{-1} r_t$$
$$= (A_t + G_t(A_t^\top)^{-1}Q_{t-1})h_{t-1} - G_t(A_t^\top)^{-1}\lambda_{t-1} - g_t,$$

for $2 \le t \le t + T$, where we denote $G_t = B_t R_t^{-1} B_t^\top, g_t = B_t R_t^{-1} r_t$ to avoid notational clusters.

We define $Q_0 = 0$, and by Eq. (33), we can derive a similar recurrent form for the initial step:

$$\lambda_1 = (A_1^\top)^{-1}\lambda_0 = (A_1^\top)^{-1}(\lambda_0 - Q_0 h_0),$$

and

$$h_1 = A_1 h_0 + B_1 u_1 = A_1 h_0 - G_1 \lambda_1 - g_1 = A_1 h_0 - G_1(A_1)^{-1}\lambda_0 - g_1$$
$$= (A_1 + G_1(A_1)^{-1}Q_0)h_0 + G_1(A_1)^{-1}\lambda_0 - g_1.$$

After rewriting above equations in matrix form, we obtain the joint update rule for $[h_t, \lambda_t]$ as desired. $\qquad\square$

Second, we show the reverse symplectic iteration theorem that is useful to compute the initial co-state $\boldsymbol{\lambda}_0$ together with the optimal first-step action. The result below directly proves Theorem 3.3 in the main text.

**Theorem A.5** (Reverse symplectic iteration). *Given LQR system in Eq. (15) with parameters $\{(\boldsymbol{A}_t, \boldsymbol{B}_t, \boldsymbol{Q}_t, \boldsymbol{R}_t, \boldsymbol{r}_t)\}_{t=1}^T$ and initial state $\boldsymbol{h}_{init}$. The optimal first-step control:*

$$\boldsymbol{u}_1 = -\boldsymbol{R}_1^{-1}(\boldsymbol{B}_1^\top (\boldsymbol{A}_1^\top)^{-1}\boldsymbol{Y}_1^{-1}(\boldsymbol{Y}_2 \boldsymbol{h}_{init} + \boldsymbol{y}_3) + \boldsymbol{r}_1), \qquad \boldsymbol{\lambda}_0 = \boldsymbol{Y}_1^{-1}(\boldsymbol{Y}_2 \boldsymbol{h}_{init} + \boldsymbol{y}_3)$$

*where* [3]

$$\begin{bmatrix} \boldsymbol{Y}_1 & \boldsymbol{Y}_2 \end{bmatrix} = \begin{bmatrix} \boldsymbol{I} & \boldsymbol{Q}_T \end{bmatrix} \overleftarrow{\prod_{t=1}^T} \boldsymbol{\Sigma}_t, \qquad \boldsymbol{y}_3 = \begin{bmatrix} \boldsymbol{I} & \boldsymbol{Q}_T \end{bmatrix} \left( \sum_{t=1}^T \left( \overleftarrow{\prod_{r=t+1}^T} \boldsymbol{\Sigma}_t \right) \boldsymbol{\beta}_t \right),$$

$$\boldsymbol{\Sigma}_t = \begin{bmatrix} (\boldsymbol{A}_t^\top)^{-1} & (\boldsymbol{A}_t^\top)^{-1}\boldsymbol{Q}_{t-1} \\ \boldsymbol{G}_t(\boldsymbol{A}_t^\top)^{-1} & \boldsymbol{A}_t + \boldsymbol{G}_t(\boldsymbol{A}_t^\top)^{-1}\boldsymbol{Q}_{t-1} \end{bmatrix}, \qquad \boldsymbol{\beta}_t = \begin{bmatrix} \boldsymbol{0} \\ -\boldsymbol{B}_t\boldsymbol{R}_t^{-1}\boldsymbol{r}_t \end{bmatrix}, \qquad \boldsymbol{G}_t = \boldsymbol{B}_t\boldsymbol{R}_t^{-1}\boldsymbol{B}_t^\top,$$

*Proof.* By Theorem A.4, the relation between $\boldsymbol{h}_0$ and $\boldsymbol{\lambda}_0$ can be found by the following "shooting" system:

$$\begin{bmatrix} \boldsymbol{h}_T \\ \boldsymbol{\lambda}_T \end{bmatrix} = \left( \overleftarrow{\prod_{t=1}^T} \boldsymbol{S}_t \right) \begin{bmatrix} \boldsymbol{h}_0 \\ \boldsymbol{\lambda}_0 \end{bmatrix} + \sum_{t=1}^T \left( \overleftarrow{\prod_{r=t+1}^T} \boldsymbol{S}_r \right) \boldsymbol{b}_t, \qquad \boldsymbol{Q}_T\boldsymbol{h}_T = \boldsymbol{\lambda}_T. \tag{34}$$

Now we define: $\boldsymbol{\Psi} = \begin{bmatrix} \boldsymbol{\Psi}_{11} & \boldsymbol{\Psi}_{12} \\ \boldsymbol{\Psi}_{21} & \boldsymbol{\Psi}_{22} \end{bmatrix} = \overleftarrow{\prod}_{t=1}^T \boldsymbol{S}_t$, and $\boldsymbol{\psi} = [\boldsymbol{\psi}_1^\top, \boldsymbol{\psi}_2^\top]^\top = \sum_{t=1}^T \left( \overleftarrow{\prod}_{r=t+1}^T \boldsymbol{S}_r \right) \boldsymbol{b}_t$. The above system can be rewritten as:

$$0 = \boldsymbol{Q}_T(\boldsymbol{\Psi}_{11}\boldsymbol{h}_0 + \boldsymbol{\Psi}_{12}\boldsymbol{\lambda}_0 + \boldsymbol{\psi}_1) - (\boldsymbol{\Psi}_{21}\boldsymbol{h}_0 + \boldsymbol{\Psi}_{22}\boldsymbol{\lambda}_0 + \boldsymbol{\psi}_2)$$
$$= (\boldsymbol{Q}_T\boldsymbol{\Psi}_{11} - \boldsymbol{\Psi}_{21})\boldsymbol{h}_0 + \boldsymbol{Q}_T\boldsymbol{\psi}_1 - \boldsymbol{\psi}_2 + (\boldsymbol{Q}_T\boldsymbol{\Psi}_{12} - \boldsymbol{\Psi}_{22})\boldsymbol{\lambda}_0.$$

This shows: $\boldsymbol{\lambda}_0 = -(\boldsymbol{Q}_T\boldsymbol{\Psi}_{12} - \boldsymbol{\Psi}_{22})^{-1}(\boldsymbol{Q}_T\boldsymbol{\Psi}_{11} - \boldsymbol{\Psi}_{21})\boldsymbol{h}_0 + (\boldsymbol{Q}_T\boldsymbol{\Psi}_{12} - \boldsymbol{\Psi}_{22})^{-1}(\boldsymbol{Q}_T\boldsymbol{\psi}_1 - \boldsymbol{\psi}_2)$. In fact, there is no need to compute every block of $\boldsymbol{\Psi}$. Instead, we can show that: $\boldsymbol{\lambda}_0 = \boldsymbol{Y}_1^{-1}(\boldsymbol{Y}_2\boldsymbol{h}_0 + \boldsymbol{y}_3)$ where:

$$\begin{bmatrix} \boldsymbol{Y}_1 & \boldsymbol{Y}_2 \end{bmatrix} = \begin{bmatrix} \boldsymbol{I} & \boldsymbol{Q}_T \end{bmatrix} \begin{bmatrix} \boldsymbol{\Psi}_{22} & -\boldsymbol{\Psi}_{21} \\ -\boldsymbol{\Psi}_{12} & \boldsymbol{\Psi}_{11} \end{bmatrix} = \begin{bmatrix} \boldsymbol{I} & \boldsymbol{Q}_T \end{bmatrix} (\boldsymbol{\Psi}^{-1})^\top$$

$$= \begin{bmatrix} \boldsymbol{I} & \boldsymbol{Q}_T \end{bmatrix} \left( \left( \overleftarrow{\prod_{t=1}^T} \boldsymbol{S}_t \right)^{-1} \right)^\top = \begin{bmatrix} \boldsymbol{I} & \boldsymbol{Q}_T \end{bmatrix} \left( \overrightarrow{\prod_{t=1}^T} \boldsymbol{S}_{T-t+1}^{-1} \right)^\top$$

$$= \begin{bmatrix} \boldsymbol{I} & \boldsymbol{Q}_T \end{bmatrix} \overleftarrow{\prod_{t=1}^T} (\boldsymbol{S}_t^{-1})^\top,$$

The second equality are due to Lemma A.6: $\boldsymbol{S}_t$ is symplectic for every $1 \leq t \leq t + T$, so is their product. Similarly, we can

---

[3]Note that, different from conventional notation, we denote $\overleftarrow{\prod}_{t=1}^T \boldsymbol{\Sigma}_t = \boldsymbol{\Sigma}_T\boldsymbol{\Sigma}_{T-1}\cdots\boldsymbol{\Sigma}_2\boldsymbol{\Sigma}_1$ to simplify subscripts, where matrices are cumulatively multiplied to the left (instead of right).

simplify the calculation of $y_3$ as below:

$$y_3 = \begin{bmatrix} I & Q_T \end{bmatrix} \begin{bmatrix} -\psi_2 \\ \psi_1 \end{bmatrix} = \begin{bmatrix} I & Q_T \end{bmatrix} J^\top \psi$$

$$= \begin{bmatrix} I & Q_T \end{bmatrix} J^\top \left( \sum_{t=1}^{T} \left( \overleftarrow{\prod_{r=t+1}^{T}} S_r \right) b_t \right)$$

$$= \begin{bmatrix} I & Q_T \end{bmatrix} \left( \sum_{t=1}^{T} \left[ \left( \overleftarrow{\prod_{r=t+1}^{T}} S_r \right)^{-1} \right]^\top J^\top b_t \right)$$

$$= \begin{bmatrix} I & Q_T \end{bmatrix} \left( \sum_{t=1}^{T} \left( \overleftarrow{\prod_{r=t+1}^{T}} (S_r^{-1})^\top \right) J^\top b_t \right),$$

where $J = \begin{bmatrix} 0 & I \\ -I & 0 \end{bmatrix}$ is the standard symplectic matrix. The fourth equality is due to symplecticity.

We let $\beta_t = J^\top b_t = [0^\top, -g_t^\top]^\top$. Using symplecticity again, we can have the closed form of $(S_t^{-1})^\top$:

$$\Sigma_t := (S_t^{-1})^\top = \begin{bmatrix} (A_t^\top)^{-1} & (A_t^\top)^{-1} Q_{t-1} \\ G_t(A_t^\top)^{-1} & A_t + G_t(A_t^\top)^{-1} Q_{t-1} \end{bmatrix}.$$

After solving $\lambda_0$ with $h_0 = h_{init}$, we can obtain $u_1$ by the KKT condition Eq. (27):

$$u_1 = -R_1^{-1}(B_1^\top \lambda_1 + r_1) = -R_1^{-1}(B_1^\top (A_1^\top)^{-1} \lambda_0 + r_1).$$

$\square$

**Lemma A.6.** $S_t$ *is a symplectic matrix:* $-JS_t^\top J = S_t^{-1}$, *where* $J = \begin{bmatrix} 0 & I \\ -I & 0 \end{bmatrix}$.

*Proof.* Consider a matrix of the following form (with subscripts removed):

$$S = \begin{bmatrix} A + G(A^\top)^{-1}Q & -G(A^\top)^{-1} \\ -(A^\top)^{-1}Q & (A^\top)^{-1} \end{bmatrix} = \begin{bmatrix} A_s & B_s \\ C_s & D_s \end{bmatrix},$$

where $A$ is invertible and $Q^\top = Q, G^\top = G$.

$S$ being symplectic is equivalent to say $S^\top J S = J$. This is further equivalent to the blockwise conditions:

$$A_s^\top C_s = C_s^\top A_s, \qquad B_s^\top D_s = D_s^\top B_s, \qquad A_s^\top D_s - C_s^\top B_s = I.$$

Then we verify that $A_s^\top C_s = C_s^\top A_s$:

$$A_s^\top C_s = (A^\top + QA^{-1}G)(-(A^\top)^{-1}Q) = -A^\top(A^\top)^{-1}Q - QA^{-1}G(A^\top)^{-1}Q = -Q - QA^{-1}G(A^\top)^{-1}Q$$

$$= -QA^{-1}A - QA^{-1}G(A^\top)^{-1}Q = (-QA^{-1})(A + G(A^\top)^{-1}Q) = C_s^\top A_s,$$

using the fact that $Q$ and $G$ are symmetric. Similarly, $B_s^\top D_s = D_s^\top B_s$ due to associativity:

$$B_s^\top D_s = (-A^{-1}G)(A^\top)^{-1} = -A^{-1}G(A^\top)^{-1} = D_s^\top B_s.$$

The last step is to verify $A_s^\top D_s - C_s^\top B_s = I$:

$$A_s^\top D_s = (A^\top + QA^{-1}G)(A^\top)^{-1} = I + QA^{-1}G(A^\top)^{-1},$$

$$C_s^\top B_s = (-QA^{-1})(-G(A^\top)^{-1}) = QA^{-1}G(A^\top)^{-1}.$$

Therefore, $A_s^\top D_s - C_s^\top B_s = I$.

$\square$

---

**Algorithm 3** Finite-horizon LQR via symplectic iteration

---

**Require:** $\{(\boldsymbol{A}_t, \boldsymbol{B}_t, \boldsymbol{Q}_t, \boldsymbol{R}_t, \boldsymbol{r}_t)\}_{t=1}^T$, initial state $\boldsymbol{h}_{init}$
**Ensure:** actions $\{\boldsymbol{u}_t\}_{t=1}^T$, states $\{\boldsymbol{h}_t\}_{t=1}^T$, co-states $\{\boldsymbol{\lambda}_t\}_{t=0}^T$
 1: Initialize $\begin{bmatrix} \boldsymbol{Y}_1 & \boldsymbol{Y}_2 \end{bmatrix} \leftarrow \begin{bmatrix} \boldsymbol{I} & \boldsymbol{Q}_T \end{bmatrix}$, $\boldsymbol{y}_3 \leftarrow \boldsymbol{0}$.
 2: **for** $t = T, \dots, 1$ **do**                                                                                                        ▷ reverse symplectic iteration
 3:    $\quad \boldsymbol{\Sigma}_t \leftarrow \begin{bmatrix} (\boldsymbol{A}_t^\top)^{-1} & (\boldsymbol{A}_t^\top)^{-1}\boldsymbol{Q}_{t-1} \\ \boldsymbol{G}_t(\boldsymbol{A}_t^\top)^{-1} & \boldsymbol{A}_t + \boldsymbol{G}_t(\boldsymbol{A}_t^\top)^{-1}\boldsymbol{Q}_{t-1} \end{bmatrix}$
 4:    $\quad \begin{bmatrix} \boldsymbol{Y}_1 & \boldsymbol{Y}_2 \end{bmatrix} \leftarrow \begin{bmatrix} \boldsymbol{Y}_1 & \boldsymbol{Y}_2 \end{bmatrix} \boldsymbol{\Sigma}_t$
 5:    $\quad \boldsymbol{y}_3 \leftarrow \boldsymbol{\Sigma}_t \boldsymbol{y}_3 + \begin{bmatrix} \boldsymbol{0} \\ -\boldsymbol{B}_t \boldsymbol{R}_t^{-1} \boldsymbol{r}_t \end{bmatrix}$
 6: **end for**
 7: $\boldsymbol{y}_3 \leftarrow \begin{bmatrix} \boldsymbol{I} & \boldsymbol{Q}_T \end{bmatrix} \boldsymbol{y}_3$
 8: Assign initial state $\boldsymbol{h}_0 \leftarrow \boldsymbol{h}_{init}$
 9: Compute initial co-state $\boldsymbol{\lambda}_0 \leftarrow \boldsymbol{Y}_1^{-1}(\boldsymbol{Y}_2 \boldsymbol{h}_{init} + \boldsymbol{y}_3)$
10: **for** $t = 1, 2, \dots, T$ **do**                                                                                                        ▷ forward symplectic iteration
11:    $\quad \begin{bmatrix} \boldsymbol{h}_t \\ \boldsymbol{\lambda}_t \end{bmatrix} \leftarrow \begin{bmatrix} \boldsymbol{A}_t + \boldsymbol{G}_t(\boldsymbol{A}_t^\top)^{-1}\boldsymbol{Q}_{t-1} & -\boldsymbol{G}_t(\boldsymbol{A}_t^\top)^{-1} \\ -(\boldsymbol{A}_t^\top)^{-1}\boldsymbol{Q}_{t-1} & (\boldsymbol{A}_t^\top)^{-1} \end{bmatrix} \begin{bmatrix} \boldsymbol{h}_{t-1} \\ \boldsymbol{\lambda}_{t-1} \end{bmatrix} + \begin{bmatrix} -\boldsymbol{B}_t \boldsymbol{R}_t^{-1} \boldsymbol{r}_t \\ \boldsymbol{0} \end{bmatrix}$
12:    $\quad \boldsymbol{u}_t \leftarrow -\boldsymbol{R}_t^{-1}(\boldsymbol{B}_t^\top \boldsymbol{\lambda}_t + \boldsymbol{r}_t)$
13: **end for**
14: **return** $\{\boldsymbol{u}_t\}_{t=1}^T$, $\{\boldsymbol{h}_t\}_{t=1}^T$, $\{\boldsymbol{\lambda}_t\}_{t=0}^T$.

---

*Remark* A.7 (Complexity analysis). Solving LQR with either Riccati or symplectic iteration has the same theoretical computational complexity $O(Td^3)$ and memory complexity $O(d^2)$.

Theorems A.4 and A.5 lead to a new algorithm for solving LQRs, as illustrated in Algorithm 3.

*Remark* A.8. Solving LQR with Algorithm 3 only requires one reverse and one forward iteration to solve all $\{(\boldsymbol{h}_t, \boldsymbol{\lambda}_t, \boldsymbol{u}_t)\}_{t=1}^T$ and $\boldsymbol{\lambda}_0$ while Algorithm 1 and 2 need one forward iteration and two reverse iterations.

## A.4. Differentiation through KKT Conditions

Below, we formally prove Theorem 3.2. The proof technique is based on Amos & Kolter (2017).

**Theorem A.9.** *Consider a TTC layer with parameters $\{(\boldsymbol{A}_t, \boldsymbol{B}_t, \boldsymbol{Q}_t, \boldsymbol{R}_t)\}_{t=1}^T)$ and initial state $\boldsymbol{h}_{init}$, suppose we have output $\boldsymbol{u}_{out} = \text{TTC}(\boldsymbol{h}_{init}, \{(\boldsymbol{A}_t, \boldsymbol{B}_t, \boldsymbol{Q}_t, \boldsymbol{R}_t)\}_{t=1}^T)$, the loss is computed as $\ell(\boldsymbol{u}_{out})$, and we have obtained $\nabla_{\boldsymbol{u}_{out}}\ell$. Then the gradients w.r.t. $\{(\boldsymbol{A}_t, \boldsymbol{B}_t, \boldsymbol{Q}_t, \boldsymbol{R}_t)\}$ are:*

$$\nabla_{\boldsymbol{A}_t}\ell = \boldsymbol{\lambda}_t \widetilde{\boldsymbol{h}}_{t-1}^\top + \widetilde{\boldsymbol{\lambda}}_t \boldsymbol{h}_{t-1}^\top, \qquad\qquad \nabla_{\boldsymbol{B}_t}\ell = \boldsymbol{\lambda}_t \widetilde{\boldsymbol{u}}_t^\top + \widetilde{\boldsymbol{\lambda}}_t \boldsymbol{u}_t^\top,$$

$$\nabla_{\boldsymbol{Q}_t}\ell = \frac{1}{2}(\boldsymbol{h}_t \widetilde{\boldsymbol{h}}_t^\top + \widetilde{\boldsymbol{h}}_t \boldsymbol{h}_t^\top), \qquad\qquad \nabla_{\boldsymbol{R}_t}\ell = \frac{1}{2}(\boldsymbol{u}_t \widetilde{\boldsymbol{u}}_t^\top + \widetilde{\boldsymbol{u}}_t \boldsymbol{u}_t^\top),$$

*and gradient w.r.t. $\boldsymbol{h}_{init}$ can be computed by $\nabla_{\boldsymbol{h}_{init}}\ell = \widetilde{\boldsymbol{\lambda}}_0$, where $\{(\boldsymbol{h}_0, \boldsymbol{\lambda}_0)\} \cup \{(\boldsymbol{h}_t, \boldsymbol{\lambda}_t, \boldsymbol{u}_t)\}_{t=1}^T$ are the solution to the KKT system associated with Eq. (8) and $\{(\widetilde{\boldsymbol{h}}_0, \widetilde{\boldsymbol{\lambda}}_0)\} \cup \{(\widetilde{\boldsymbol{h}}_t, \widetilde{\boldsymbol{\lambda}}_t, \widetilde{\boldsymbol{u}}_t)\}_{t=1}^T$ are the solution to the KKT system corresponding to the following LQR system with $\widetilde{\boldsymbol{h}}_0 = \boldsymbol{0}$:*

$$\min \frac{1}{2}\sum_{t=1}^T (\widetilde{\boldsymbol{h}}_t^\top \boldsymbol{Q}_t \widetilde{\boldsymbol{h}}_t + \widetilde{\boldsymbol{u}}_t^\top \boldsymbol{R}_t \widetilde{\boldsymbol{u}}_t) + \nabla_{\boldsymbol{u}_{out}}\ell^\top \widetilde{\boldsymbol{u}}_1 \tag{35}$$

$$s.t. \quad \widetilde{\boldsymbol{h}}_t = \boldsymbol{A}_t \widetilde{\boldsymbol{h}}_{t-1} + \boldsymbol{B}_t \widetilde{\boldsymbol{u}}_t, \quad 1 \le t \le T. \tag{36}$$

*Proof.* To see this more easily, we consider a matrix form of the KKT system in Eq. (15):

$$L = \frac{1}{2}\boldsymbol{x}^\top \boldsymbol{C}\boldsymbol{x} + \boldsymbol{\xi}^\top(\boldsymbol{F}\boldsymbol{x} - \boldsymbol{b})$$

where we concatenate all states and actions in $\boldsymbol{x} = [\boldsymbol{h}_0^\top, \cdots, \boldsymbol{h}_T^\top, \boldsymbol{u}_1, \cdots, \boldsymbol{u}_T^\top]^\top \in \mathbb{R}^{(2T+1)d}$, all co-states in $\boldsymbol{\xi} = [\boldsymbol{\lambda}_0^\top, \cdots, \boldsymbol{\lambda}_T^\top]^\top \in \mathbb{R}^{(T+1)d}$, and define cost matrix $\boldsymbol{C} \in \mathbb{R}^{(2T+1)d \times (2T+1)d}$ and linear constraints $\boldsymbol{F} \in \mathbb{R}^{(T+1)d \times (2T+1)d}$ as below[4]:

$$
\boldsymbol{C} = \begin{bmatrix} \boldsymbol{0} & & & & \\ & \frac{\boldsymbol{Q}_1 + \boldsymbol{Q}_1^\top}{2} & & & \\ & & \ddots & & \\ & & & \frac{\boldsymbol{R}_1 + \boldsymbol{R}_1^\top}{2} & \\ & & & & \ddots \end{bmatrix}, \quad
\boldsymbol{F} = \left[ \begin{array}{ccccc|cccc} -\boldsymbol{I} & \boldsymbol{0} & \boldsymbol{0} & \cdots & \boldsymbol{0} & \boldsymbol{0} & \boldsymbol{0} & \cdots & \boldsymbol{0} \\ \boldsymbol{A}_1 & -\boldsymbol{I} & \boldsymbol{0} & \cdots & \boldsymbol{0} & \boldsymbol{B}_1 & \boldsymbol{0} & \cdots & \boldsymbol{0} \\ \boldsymbol{0} & \boldsymbol{A}_2 & -\boldsymbol{I} & \cdots & \boldsymbol{0} & \boldsymbol{0} & \boldsymbol{B}_2 & \cdots & \boldsymbol{0} \\ \vdots & \vdots & \ddots & \ddots & \vdots & \vdots & \vdots & \ddots & \vdots \\ \boldsymbol{0} & \boldsymbol{0} & \cdots & \boldsymbol{A}_T & -\boldsymbol{I} & \boldsymbol{0} & \boldsymbol{0} & \cdots & \boldsymbol{B}_T \end{array} \right], \quad
\boldsymbol{b} = \begin{bmatrix} -\boldsymbol{h}_{init} \\ \boldsymbol{0} \\ \boldsymbol{0} \\ \vdots \\ \boldsymbol{0} \end{bmatrix}.
$$

The Eqs. (24)-(29) can be expressed as:

$$
\nabla_{\boldsymbol{x}} L = \boldsymbol{C}\boldsymbol{x} + \boldsymbol{F}^\top \boldsymbol{\xi} = \boldsymbol{0}, \qquad\qquad \nabla_{\boldsymbol{\xi}} L = \boldsymbol{F}\boldsymbol{x} - \boldsymbol{b} = \boldsymbol{0}.
$$

Consider a scalar parameter $\theta$ from any of $\{(\boldsymbol{A}_t, \boldsymbol{B}_t, \boldsymbol{Q}_t, \boldsymbol{R}_t)\}_{t=1}^T$ or $\boldsymbol{h}_{init}$, we take derivatives w.r.t. $\theta$ from both sides of the above equations:

$$
\frac{\partial}{\partial \theta} \boldsymbol{C}\boldsymbol{x} + \boldsymbol{C}\frac{\partial}{\partial \theta}\boldsymbol{x} + \frac{\partial}{\partial \theta}\boldsymbol{F}^\top \boldsymbol{\xi} + \boldsymbol{F}^\top \frac{\partial}{\partial \theta}\boldsymbol{\xi} = \boldsymbol{0}, \qquad\qquad \frac{\partial}{\partial \theta}\boldsymbol{F}\boldsymbol{x} + \boldsymbol{F}\frac{\partial}{\partial \theta}\boldsymbol{x} - \frac{\partial}{\partial \theta}\boldsymbol{b} = \boldsymbol{0},
$$

which can be rewritten in the following matrix form:

$$
\begin{bmatrix} \boldsymbol{C} & \boldsymbol{F}^\top \\ \boldsymbol{F} & \boldsymbol{0} \end{bmatrix} \begin{bmatrix} \frac{\partial}{\partial \theta}\boldsymbol{x} \\ \frac{\partial}{\partial \theta}\boldsymbol{\xi} \end{bmatrix} = \begin{bmatrix} -\frac{\partial}{\partial \theta}\boldsymbol{C}\boldsymbol{x} - \frac{\partial}{\partial \theta}\boldsymbol{F}^\top \boldsymbol{\xi} \\ -\frac{\partial}{\partial \theta}\boldsymbol{F}\boldsymbol{x} + \frac{\partial}{\partial \theta}\boldsymbol{b} \end{bmatrix}. \tag{37}
$$

This implies that we can solve the following equation to obtain $\left[ \frac{\partial}{\partial \theta}\boldsymbol{x}^\top, \frac{\partial}{\partial \theta}\boldsymbol{\xi}^\top \right]^\top$:

$$
\begin{bmatrix} \frac{\partial}{\partial \theta}\boldsymbol{x} \\ \frac{\partial}{\partial \theta}\boldsymbol{\xi} \end{bmatrix} = - \begin{bmatrix} \boldsymbol{C} & \boldsymbol{F}^\top \\ \boldsymbol{F} & \boldsymbol{0} \end{bmatrix}^{-1} \begin{bmatrix} \frac{\partial}{\partial \theta}\boldsymbol{C}\boldsymbol{x} + \frac{\partial}{\partial \theta}\boldsymbol{F}^\top \boldsymbol{\xi} \\ \frac{\partial}{\partial \theta}\boldsymbol{F}\boldsymbol{x} - \frac{\partial}{\partial \theta}\boldsymbol{b} \end{bmatrix}.
$$

Note that our goal is to obtain $\frac{\partial}{\partial \theta}\ell$. By chain rule, we obtain:

$$
\frac{\partial}{\partial \theta}\ell = \begin{bmatrix} \nabla_{\boldsymbol{x}}\ell \\ \nabla_{\boldsymbol{\xi}}\ell \end{bmatrix}^\top \begin{bmatrix} \frac{\partial}{\partial \theta}\boldsymbol{x} \\ \frac{\partial}{\partial \theta}\boldsymbol{\xi} \end{bmatrix} = - \underbrace{\begin{bmatrix} \nabla_{\boldsymbol{x}}\ell \\ \nabla_{\boldsymbol{\xi}}\ell \end{bmatrix}^\top \begin{bmatrix} \boldsymbol{C} & \boldsymbol{F}^\top \\ \boldsymbol{F} & \boldsymbol{0} \end{bmatrix}^{-1}}_{\begin{bmatrix} \widetilde{\boldsymbol{x}}^\top & \widetilde{\boldsymbol{\xi}}^\top \end{bmatrix}} \underbrace{\begin{bmatrix} \frac{\partial}{\partial \theta}\boldsymbol{C}\boldsymbol{x} + \frac{\partial}{\partial \theta}\boldsymbol{F}^\top \boldsymbol{\xi} \\ \frac{\partial}{\partial \theta}\boldsymbol{F}\boldsymbol{x} - \frac{\partial}{\partial \theta}\boldsymbol{b} \end{bmatrix}}_{d_\theta},
$$

where we group the left two terms as $\begin{bmatrix} \widetilde{\boldsymbol{x}}^\top, \widetilde{\boldsymbol{\xi}}^\top \end{bmatrix} = \begin{bmatrix} -\nabla_{\boldsymbol{x}}\ell \\ -\nabla_{\boldsymbol{\xi}}\ell \end{bmatrix}^\top \begin{bmatrix} \boldsymbol{C} & \boldsymbol{F}^\top \\ \boldsymbol{F} & \boldsymbol{0} \end{bmatrix}^{-1}$ which are independent of the choice of $\theta$ and

denote the remaining $\theta$-dependent term as $d_\theta = \begin{bmatrix} \frac{\partial}{\partial \theta}\boldsymbol{C}\boldsymbol{x} + \frac{\partial}{\partial \theta}\boldsymbol{F}^\top \boldsymbol{\xi} \\ \frac{\partial}{\partial \theta}\boldsymbol{F}\boldsymbol{x} - \frac{\partial}{\partial \theta}\boldsymbol{b} \end{bmatrix}$.

First, $\begin{bmatrix} \widetilde{\boldsymbol{x}}^\top, \widetilde{\boldsymbol{\xi}}^\top \end{bmatrix}$ is the solution to the following system:

$$
\begin{bmatrix} \boldsymbol{C} & \boldsymbol{F}^\top \\ \boldsymbol{F} & \boldsymbol{0} \end{bmatrix} \begin{bmatrix} \widetilde{\boldsymbol{x}} \\ \widetilde{\boldsymbol{\xi}} \end{bmatrix} = \begin{bmatrix} -\nabla_{\boldsymbol{x}}\ell \\ -\nabla_{\boldsymbol{\xi}}\ell \end{bmatrix},
$$

which corresponds to the KKT condition of the following Lagrangian function:

$$
\widetilde{L} = \frac{1}{2}\widetilde{\boldsymbol{x}}^\top \boldsymbol{C}\widetilde{\boldsymbol{x}} + \nabla_{\boldsymbol{x}}\ell^\top \widetilde{\boldsymbol{x}} + \widetilde{\boldsymbol{\xi}}^\top (\boldsymbol{F}\widetilde{\boldsymbol{x}} + \nabla_{\boldsymbol{\xi}}\ell). \tag{38}
$$

---

[4] Note that although flipping signs in $\boldsymbol{F}$ and $\boldsymbol{b}$ yields an equivalent system, we choose the sign convention to be consistent with Eq. (22) and the KKT conditions in Eqs. (24)-(29) so that the signs of the optimal $\boldsymbol{\lambda}_t$'s for both systems are aligned.

Since $\ell$ is only computed from $\boldsymbol{u}_1$, $\nabla_{\boldsymbol{\xi}}\ell = \boldsymbol{0}$ and $\nabla_{\boldsymbol{x}}\ell^\top = [\boldsymbol{0}_{(T+1)d}^\top, \nabla_{\boldsymbol{u}_{out}}\ell^\top, \boldsymbol{0}_{(T-1)d}^\top]$. By rewriting $\widetilde{\boldsymbol{x}} = [\widetilde{\boldsymbol{h}}_0^\top, \cdots, \widetilde{\boldsymbol{h}}_T^\top, \widetilde{\boldsymbol{u}}_1^\top, \cdots, \widetilde{\boldsymbol{u}}_T^\top]^\top$ and $\widetilde{\boldsymbol{\xi}} = [\widetilde{\boldsymbol{\lambda}}_0^\top, \cdots, \widetilde{\boldsymbol{\lambda}}_T^\top]^\top$, the Lagrangian function in Eq. (38) corresponds to the LQR problem in Eq. (35).

Now we analyze $\boldsymbol{d}_\theta$. Suppose $\theta := \boldsymbol{A}_{t,ij}$, then $\frac{\partial}{\partial\theta}\boldsymbol{C} = \boldsymbol{0}$, $\frac{\partial}{\partial\theta}\boldsymbol{b} = \boldsymbol{0}$, and $\left[\frac{\partial}{\partial\theta}\boldsymbol{F}\right]_{td+i,(t-1)d+j} = 1$ with all other entries being zero. This gives

$$\frac{\partial\ell}{\partial\boldsymbol{A}_{t,ij}} = \begin{bmatrix}\widetilde{\boldsymbol{x}}^\top & \widetilde{\boldsymbol{\xi}}^\top\end{bmatrix}\begin{bmatrix}\frac{\partial}{\partial\theta}\boldsymbol{F}^\top\boldsymbol{\xi} \\ \frac{\partial}{\partial\theta}\boldsymbol{F}\boldsymbol{x}\end{bmatrix} = \boldsymbol{\xi}_{td+i}\widetilde{\boldsymbol{x}}_{(t-1)d+j} + \widetilde{\boldsymbol{\xi}}_{td+i}\boldsymbol{x}_{(t-1)d+j}.$$

Hence $\frac{\partial}{\partial\boldsymbol{A}_t}\ell = \boldsymbol{\lambda}_t\widetilde{\boldsymbol{h}}_{t-1}^\top + \widetilde{\boldsymbol{\lambda}}_t\boldsymbol{h}_{t-1}^\top$.

Suppose $\theta := \boldsymbol{B}_{t,ij}$, then $\frac{\partial}{\partial\theta}\boldsymbol{C} = \boldsymbol{0}$, $\frac{\partial}{\partial\theta}\boldsymbol{b} = \boldsymbol{0}$, and $\left[\frac{\partial}{\partial\theta}\boldsymbol{F}\right]_{td+i,(T+t)d+j} = 1$. Similarly, we have

$$\frac{\partial\ell}{\partial\boldsymbol{B}_{t,ij}} = \boldsymbol{\xi}_{td+i}\widetilde{\boldsymbol{x}}_{(T+t)d+j} + \widetilde{\boldsymbol{\xi}}_{td+i}\boldsymbol{x}_{(T+t)d+j}$$

In matrix form, it is equivalent to $\frac{\partial}{\partial\boldsymbol{B}_t}\ell = \boldsymbol{\lambda}_t\widetilde{\boldsymbol{u}}_t^\top + \widetilde{\boldsymbol{\lambda}}_t\boldsymbol{u}_t^\top$.

Likewise, consider $\theta := \boldsymbol{Q}_{t,ij}$ (or $\theta := \boldsymbol{R}_{t,ij}$), then $\frac{\partial}{\partial\theta}\boldsymbol{F} = \boldsymbol{0}$, $\frac{\partial}{\partial\theta}\boldsymbol{b} = \boldsymbol{0}$, and $\left[\frac{\partial}{\partial\theta}\boldsymbol{C}\right]_{td+i,td+j} = \left[\frac{\partial}{\partial\theta}\boldsymbol{C}\right]_{td+j,td+i} = 1/2$ (or $\left[\frac{\partial}{\partial\theta}\boldsymbol{C}\right]_{(T+t)d+i,(T+t)d+j} = \left[\frac{\partial}{\partial\theta}\boldsymbol{C}\right]_{(T+t)d+j,(T+t)d+i} = 1/2$). Henceforth, we can derive:

$$\frac{\partial\ell}{\partial\boldsymbol{Q}_{t,ij}} = \frac{1}{2}(\boldsymbol{x}_{td+i}\widetilde{\boldsymbol{x}}_{td+j} + \widetilde{\boldsymbol{x}}_{td+i}\boldsymbol{x}_{td+j}), \qquad \frac{\partial\ell}{\partial\boldsymbol{R}_{t,ij}} = \frac{1}{2}(\boldsymbol{x}_{(T+t)d+i}\widetilde{\boldsymbol{x}}_{(T+t)d+j} + \widetilde{\boldsymbol{x}}_{(T+t)d+i}\boldsymbol{x}_{(T+t)d+j}),$$

i.e., $\frac{\partial}{\partial\boldsymbol{Q}_t}\ell = \frac{1}{2}(\boldsymbol{h}\widetilde{\boldsymbol{h}}_t^\top + \widetilde{\boldsymbol{h}}_t\boldsymbol{h}^\top)$ and $\frac{\partial}{\partial\boldsymbol{R}_t}\ell = \frac{1}{2}(\boldsymbol{u}\widetilde{\boldsymbol{u}}_t^\top + \widetilde{\boldsymbol{u}}_t\boldsymbol{u}^\top)$, respectively.

Finally, we examine $\theta := \boldsymbol{h}_{init,i}$. In this case, $\frac{\partial}{\partial\theta}\boldsymbol{C} = \boldsymbol{0}$, $\frac{\partial}{\partial\theta}\boldsymbol{F} = \boldsymbol{0}$, and $\frac{\partial}{\partial\theta}\boldsymbol{b}_i = 1$ while $\frac{\partial}{\partial\theta}\boldsymbol{b}_j = 0$ for $i \neq j$. Therefore, $\frac{\partial}{\partial\boldsymbol{h}_{init,i}}\ell = \boldsymbol{\xi}_i$, then $\frac{\partial}{\partial\boldsymbol{h}_{init}}\ell = \widetilde{\boldsymbol{\lambda}}_0$. $\qquad\square$

## B. Hardware Co-Designs and Optimization for TTC

In this section, we elaborate on the techniques to further improve the hardware efficiency of our LQR solver by taking advantage of Theorem 3.3. The pseudo-code for forward and backward passes are described in Algorithms 4 and 5.

**Kernel Fusion.** The algorithm implied by Theorem 3.3 is amenable to kernel fusion, since the dominant computation in the cumulative matrix product involves only basic tensor operations. We parallelize Eq. (12) by letting each kernel maintain a row block of $[\boldsymbol{Y}_1, \boldsymbol{Y}_2]$ and $\boldsymbol{y}_3$ on SRAM while loading $\boldsymbol{\Sigma}_t$ to perform matrix product in a loop. In fact, neither $\boldsymbol{\Sigma}_t$ nor $\boldsymbol{G}_t$ needs to be explicitly instantiated in HBM. Instead, $\boldsymbol{\Sigma}_t$ admits the following factorization:

$$\boldsymbol{\Sigma}_t = \begin{bmatrix}\boldsymbol{I} & \boldsymbol{0} \\ \boldsymbol{B}_t\boldsymbol{R}_t^{-1}\boldsymbol{B}_t^\top & \boldsymbol{I}\end{bmatrix}\begin{bmatrix}(\boldsymbol{A}_t^{-1})^\top & \boldsymbol{0} \\ \boldsymbol{0} & \boldsymbol{A}_t\end{bmatrix}\begin{bmatrix}\boldsymbol{I} & \boldsymbol{Q}_{t-1} \\ \boldsymbol{0} & \boldsymbol{I}\end{bmatrix}.$$

This allows the CUDA kernel to stream $\boldsymbol{B}_t$, $\boldsymbol{R}_t$, $\boldsymbol{A}_t$, and $\boldsymbol{Q}_{t-1}$ sequentially into on-chip SRAM to join the computation of each block in $\boldsymbol{\Sigma}_t$. By fusing all tensor operations into a single kernel, our new solver further reduces HBM access and thereby alleviates the I/O latency. Finally, once $\boldsymbol{Y}_1$, $\boldsymbol{Y}_2$, and $\boldsymbol{y}_3$ are obtained, we leverage dense linear algebra routines in the cuBLAS library to compute $\boldsymbol{Y}_1^{-1}\boldsymbol{Y}_2$ and $\boldsymbol{Y}_1^{-1}\boldsymbol{y}_3$. Through a similar approach, we are further able to implement a CUDA kernel for the forward symplectic iteration described in Algorithm 3. Note that $\boldsymbol{S}_t$ follows a similar factorized form:

$$\boldsymbol{S}_t = \begin{bmatrix}\boldsymbol{I} & -\boldsymbol{B}_t\boldsymbol{R}_t^{-1}\boldsymbol{B}_t^\top \\ \boldsymbol{0} & \boldsymbol{I}\end{bmatrix}\begin{bmatrix}\boldsymbol{A}_t & \boldsymbol{0} \\ \boldsymbol{0} & (\boldsymbol{A}_t^{-1})^\top\end{bmatrix}\begin{bmatrix}\boldsymbol{I} & \boldsymbol{0} \\ -\boldsymbol{Q}_{t-1} & \boldsymbol{I}\end{bmatrix},$$

allowing accumulating $\boldsymbol{S}_t$ via streaming $\boldsymbol{Q}_{t-1}$, $\boldsymbol{A}_t$, $\boldsymbol{R}_t$, and $\boldsymbol{B}_t$ in order. For gradient computation, we fuse two forward symplectic iterations for both primal and dual LQRs together, to efficiently obtain and combine trajectories $\{(\boldsymbol{h}_t, \boldsymbol{u}_t, \boldsymbol{\lambda}_t)\}$ and $\{(\widetilde{\boldsymbol{h}}_t, \widetilde{\boldsymbol{u}}_t, \widetilde{\boldsymbol{\lambda}}_t)\}$ on chip without instantiating them on HBM.

---

**Algorithm 4** Parallelized implementation of TTC layer's forward pass

---

**Require:** $\{(\boldsymbol{A}_t, \boldsymbol{B}_t, \boldsymbol{Q}_t, \boldsymbol{R}_t)\}_{t=1}^T$, initial state $\boldsymbol{h}_{init}$, block size $b$.
**Ensure:** First-step actions $\boldsymbol{u}_{out}$ and caches for backward pass.

1: Initialize $\boldsymbol{Y}_1 \leftarrow \boldsymbol{I}, \boldsymbol{Y}_2 \leftarrow \boldsymbol{Q}_T, \boldsymbol{Y}_3 \leftarrow \boldsymbol{0}_{d \times d}$ on HBM.
2: **for** $i = 1, \ldots, \lceil d/b \rceil$ in parallel **do**
3:      Load on chip: $\boldsymbol{Y}_1^s \leftarrow \boldsymbol{Y}_1[(i-1)b : ib, :]$.
4:      Load on chip: $\boldsymbol{Y}_2^s \leftarrow$ load $\boldsymbol{Y}_2[(i-1)b : ib, :]$.
5:      Initialize on chip: $\boldsymbol{Y}_3^s \leftarrow \boldsymbol{0}_{d \times d}$.
6:      **for** $t = T, \ldots, 1$ **do**                       $\triangleright$ backward symlectic iteration
7:          Load and compute $\boldsymbol{A}_t, \boldsymbol{B}_t, \boldsymbol{Q}_t, \boldsymbol{R}_t$ on chip.
8:          $\boldsymbol{Y}_3^s \leftarrow \mathtt{matmul}(\mathtt{matmul}(\boldsymbol{Y}_2^s, \boldsymbol{B}_t), \boldsymbol{R}_t^{-1})$
9:          $\boldsymbol{Y}_1^s \leftarrow \boldsymbol{Y}_1^s + \mathtt{matmul}(\boldsymbol{Y}_3^s, \boldsymbol{B}_t^\top)$
10:         $\boldsymbol{Y}_1^s \leftarrow \mathtt{matmul}(\boldsymbol{Y}_1^s, (\boldsymbol{A}_t^{-1})^\top)$
11:         $\boldsymbol{Y}_2^s \leftarrow \mathtt{matmul}(\boldsymbol{Y}_2^s, \boldsymbol{A}_t)$
12:         $\boldsymbol{Y}_2^s \leftarrow \boldsymbol{Y}_2^s + \mathtt{matmul}(\boldsymbol{Y}_1^s, \boldsymbol{Q}_{t-1})$
13:         $\boldsymbol{D}^s \leftarrow \max(\mathtt{norm}(\boldsymbol{Y}_1^s, \mathtt{dim}=1), \mathtt{norm}(\boldsymbol{Y}_2^s, \mathtt{dim}=1))$      $\triangleright$ row-wise normalization
14:         $\boldsymbol{Y}_1^s \leftarrow \mathtt{div}(\boldsymbol{Y}_1^s, \boldsymbol{D}^s[:, \mathtt{None}])$
15:         $\boldsymbol{Y}_2^s \leftarrow \mathtt{div}(\boldsymbol{Y}_2^s, \boldsymbol{D}^s[:, \mathtt{None}])$
16:         $\boldsymbol{Y}_3^s \leftarrow \mathtt{div}(\boldsymbol{Y}_3^s, \boldsymbol{D}^s[:, \mathtt{None}])$
17:      **end for**
18:      Write to HBM: $\boldsymbol{Y}_1[(i-1)b : ib, :] \leftarrow \boldsymbol{Y}_1^s$.
19:      Write to HBM: $\boldsymbol{Y}_2[(i-1)b : ib, :] \leftarrow \boldsymbol{Y}_2^s$.
20:      Write to HBM: $\boldsymbol{Y}_3[(i-1)b : ib, :] \leftarrow \boldsymbol{Y}_3^s$.
21: **end for**
22: $(\boldsymbol{L}, \boldsymbol{U}) \leftarrow \mathtt{lu\_factor}(\boldsymbol{Y}_1)$.
23: $\boldsymbol{P} \leftarrow \mathtt{lu\_solve}(\boldsymbol{L}, \boldsymbol{U}, \boldsymbol{Y}_2)$.
24: $\boldsymbol{\lambda}_0 \leftarrow \mathtt{matmul}(\boldsymbol{P}, \boldsymbol{h}_{init})$.
25: $\boldsymbol{u}_1 \leftarrow \mathtt{matmul}(\boldsymbol{R}_1^{-1}, \mathtt{matmul}(\boldsymbol{B}_1, \boldsymbol{\lambda}_0))$.
26: Save $(\boldsymbol{L}, \boldsymbol{U}), \boldsymbol{Y}_3, \boldsymbol{\lambda}_0$ for backward.
27: Write to HBM: $\boldsymbol{u}_{out} \leftarrow \boldsymbol{u}_1$.

---

**Numerical Stability.** Cumulative products of symplectic matrices may suffer from numerical instability, as their eigenvalues occur in reciprocal pairs (McDuff & Salamon, 2017). As a result, directly computing the symplectic iteration can cause the entries of the resulting matrices $[\boldsymbol{Y}_1, \boldsymbol{Y}_2]$ to grow exponentially, leading to numerical overflow and instability when computing $\boldsymbol{Y}_1^{-1}$. To address this issue, we introduce a normalization scheme that keeps the values in $[\boldsymbol{Y}_1, \boldsymbol{Y}_2]$ within a controlled range. We observe that row-wise preconditioning of $\boldsymbol{Y}_1$ and $\boldsymbol{Y}_2$ does not affect the final result $\boldsymbol{Y}_1^{-1} \boldsymbol{Y}_2$. Based on this observation, we construct a diagonal matrix $\boldsymbol{D}$ with entries $\boldsymbol{D}_{ii} = \max\{\|\boldsymbol{Y}_{1,i}\|_1, \|\boldsymbol{Y}_{2,i}\|_1\}$ and apply $\boldsymbol{D}^{-1}$ to $[\boldsymbol{Y}_1, \boldsymbol{Y}_2]$ immediately after each step. Although $[\boldsymbol{Y}_1, \boldsymbol{Y}_2]$ are partitioned row-wise and distributed across CUDA blocks, this normalization can be performed independently within each kernel without synchronization.

**Mixed Precision.** The TTC layer supports either full-precision (`float32`) or half-precision (`float16` or `bfloat16`) inputs. Within the fused kernels of the reverse symplectic iteration, we deliberately keep all on-chip tensors $\boldsymbol{Y}_1$ and $\boldsymbol{Y}_2$ in `float32` to preserve numerical accuracy during long-horizon matrix accumulations. The output tensor $\boldsymbol{Y}_1$ also remains in `float32`, since LU decomposition is particularly sensitive to precision, whereas other intermediate quantities can be safely cast to lower precision when written back to HBM. In the forward symplectic iteration for gradient computation, all on-chip tensors are likewise maintained in `float32` and only cast to the target precision when stored in HBM.

**Caching for Backward.** Taking a closer look at the dual LQR in Theorem 3.2, we observe that the coefficients associated with the affine terms are nonzero if and only if $t = 1$. Therefore, compared with the primal LQR, the symplectic iteration described in Theorem 3.3 differs only at the final step. This mathematical similarity allows several intermediate quantities computed during the forward pass to be reused in the backward pass: (1) The matrix $\boldsymbol{Y}_1$, required in both the primal and dual LQRs, is identical in the two cases. Hence, it can be cached after the forward symplectic iteration. In practice, we store its LU decomposition, since forward and backward computations only require applying $\boldsymbol{Y}_1^{-1}$ to $\boldsymbol{Y}_2$ and $\boldsymbol{y}_3$, respectively. (2)

---

**Algorithm 5** Parallelized implementation of TTC layer's backward pass

---

**Require:** $\{(A_t, B_t, Q_t, R_t)\}_{t=1}^T$, initial state $h_{init}$, gradient $\nabla_{u_{out}}\ell$ and caches from forward pass $(L, U)$, $Y_3$, $\lambda_0$.
**Ensure:** Gradients $\{(G_{A_t}, G_{B_t}, G_{Q_t}, G_{R_t})\}_{t=1}^T$ and $G_{h_{init}}$ on HBM.

  1: Initialize $h \leftarrow h_{init}$, $\lambda \leftarrow \lambda_0$, $\widetilde{h} \leftarrow h_{init}$ on chip.
  2: Load $(L, U)$, $Y_3$, $\lambda_0$ from foward cache.
  3: $y_3 \leftarrow \texttt{matmul}(Y_3, -\nabla_{u_{out}}\ell)$.
  4: Initialize $\widetilde{\lambda} \leftarrow \texttt{lu\_solve}(L, U, y_3)$ on chip.
  5: Write to HBM: $G_{h_{init}} \leftarrow \widetilde{\lambda}$.
  6: **for** $t = 1, \ldots, T$ **do**                                              ▷ forward symlectic iteration
  7:     Load and compute $A_t, B_t, Q_t, R_t$ on chip.
  8:     $\lambda \leftarrow \lambda - \texttt{matmul}(Q_{t-1}, h)$
  9:     $\widetilde{\lambda} \leftarrow \widetilde{\lambda} - \texttt{matmul}(Q_{t-1}, \widetilde{h})$
10:     $\lambda \leftarrow \texttt{matmul}((A_t^{-1})^\top, \lambda)$
11:     $\widetilde{\lambda} \leftarrow \texttt{matmul}((A_t^{-1})^\top, \widetilde{\lambda})$
12:     Write to HBM: $G_{A_t} \leftarrow \lambda\widetilde{h}^\top + \widetilde{\lambda}h^\top$.
13:     $u \leftarrow -\texttt{matmul}(R_t^{-1}, \texttt{matmul}(B_t^\top, \lambda))$
14:     $\widetilde{u} \leftarrow -\texttt{matmul}(R_t^{-1}, \texttt{matmul}(B_t^\top, \widetilde{\lambda}))$
15:     **if** $t = 1$ **then**
16:         $\widetilde{u} \leftarrow \widetilde{u} - \texttt{matmul}(R_t^{-1}, \nabla_{u_{out}}\ell)$
17:     **end if**
18:     Write to HBM: $G_{B_t} \leftarrow \lambda\widetilde{u}^\top + \widetilde{\lambda}u^\top$.
19:     $h \leftarrow \texttt{matmul}(A_t, h) + \texttt{matmul}(B_t, u)$
20:     $\widetilde{h} \leftarrow \texttt{matmul}(A_t, \widetilde{h}) + \texttt{matmul}(B_t, \widetilde{u})$
21:     Write to HBM: $G_{Q_t} \leftarrow (h\widetilde{h}^\top + \widetilde{h}h^\top)/2$.
22:     Write to HBM: $G_{R_t} \leftarrow (u\widetilde{u}^\top + \widetilde{u}u^\top)/2$.
23: **end for**

---

Note that $y_3 = \begin{bmatrix} I & Q_T \end{bmatrix} \left(\prod_{t=1}^{T-1} \Sigma_{T-t+1}\right) \begin{bmatrix} 0 & -B_t R_t^{-1}\nabla_{u_{out}}\ell \end{bmatrix}^\top$. The right block of $\begin{bmatrix} I & Q_T \end{bmatrix} \left(\prod_{t=1}^{T-1} \Sigma_{T-t+1}\right)$ is already computed during the forward pass. This block can be cached and directly reused in the backward computation. By exploiting these caching strategies, we eliminate the need for an additional symplectic iteration during backpropagation. For gradient computation, we fuse two forward symplectic iterations in one GPU kernel for computing and combining trajectories $\{(h_t, u_t, \lambda_t)\}$ and $\{(\widetilde{h}_t, \widetilde{u}_t, \widetilde{\lambda}_t)\}$ on chip without instantiating them on HBM. See Appendix B for more details.

## C. Architectural Designs for TTC

**Formulation of TTC Block.** As mentioned in Sec. 4, TTC-Net employs a linear mapping to project context-rich token features produced by attention into the initial state of the TTC layer. The output of the TTC layer is then normalized and passed through another linear mapping, after which it is added back to the residual stream of the LLM backbone. Please refer to Fig. 4 for a quick overview. The entire block can be expressed as:

$$h_0 = W_{in}\texttt{LN}(x_{in}), \qquad\qquad \{(A_t, B_t, Q_t, R_t)\}_{t=1}^T = \texttt{Cxt}(h_0, T),$$
$$o = \texttt{TTC}\left(h_0, \{(A_t, B_t, Q_t, R_t)\}_{t=1}^T\right), \qquad\qquad x_{out} = x_{in} + W_{out}\texttt{LN}(o),$$

where $x_{in}$ is a token feature output by the preceding attention layer, $x_{out}$ represents the final output, $\texttt{LN}$ denotes a layer normalization, $\texttt{Cxt}(\cdot)$ is the contextualization mechanism introduced above, and $W_{in}, W_{out}$ are input and output projection matrices.

**Multi-Head Structure.** Like many modern architectures, the TTC layer adopts a multi-head structure. The input state is partitioned into small blocks (e.g., of size 16), each of which generates a distinct set of TTC parameters for a corresponding head. For memory efficiency, we share the basis matrices $\{Q^{(i)}\}_{i=1}^r$ and $\{B^{(i)}\}_{i=1}^r$ as well as $s_Q$ and $s_B$ across all heads. The head-wise outputs are then concatenated and combined with a linear layer.

**Parameterization.** We parameterize the time-modulation coefficients $\mathbf{\Gamma}_\square$, where $\square \in \{A, B, Q\}$, in the logarithmic domain. This improves numerical stability and efficiency when evaluating powers of the form $\mathbf{\Gamma}_\square^t$. Specifically, we use a linear layer followed by a softplus activation, denoted $s_{\Gamma_\square}$, to generate the log-scale time-modulation coefficients. In the kernel, these are converted into time-dependent coefficients via $\exp(-t \cdot s_{\Gamma_\square}(\boldsymbol{h}_0))$. Furthermore, we observe that $\boldsymbol{R}_t$ appears only in its inverse form $\boldsymbol{R}_t^{-1}$. Therefore, rather than computing the matrix inverse at each time step, we directly let $s_R$ output the diagonal of $\boldsymbol{R}_t^{-1}$ and use it as input to the LQR solver. During the backward, we directly compute gradients in terms of log-scale time-modulation coefficients and the inverse of $\boldsymbol{R}_t$.

**Training Strategy.** The time modulation strategy described above enables the TTC layer to support an arbitrary planning horizon $T$ as input. However, training the TTC layer with a fixed horizon can induce distribution shift in downstream layers when a different horizon is used at test time. To mitigate this issue, we adopt a mixed-horizon training strategy. At each training iteration, we sample a random planning horizon from a truncated Poisson log-normal (PLN) distribution (Geiping et al., 2025): $\tau \sim \mathcal{N}\left(\log(T_\mu) - \frac{1}{2}T_\sigma^2, T_\sigma^2\right)$, $T_{train} \sim \text{Poisson}(\exp(\tau)) + 1$, where $T_\mu$ denotes the mean of $T$, and $T_\sigma$ controls the standard deviation. To ensure $T_{train}$ is bounded, we use rejection sampling: we repeatedly sample $T_{train}$ until the drawn $T_{train}$ falls within the range. In our experiments, we set the mean to $T_\mu = 8$, the log-scale standard deviation to $T_\sigma = 0.1$, and the maximum horizon to 32.

**Test-Time Scaling.** At test time, the planning horizon $T_{test}$ can be adjusted arbitrarily. Increasing $T_{test}$ causes the symplectic iteration to run longer and consume additional FLOPs; however, a longer horizon also enables the model to explore trajectories more deeply and emphasize long-term objectives, often leading to more accurate solutions. Consequently, the TTC layer can adaptively allocate additional computation to reason longer and plan over an extended horizon. This exposes a new test-time scaling (Snell et al., 2024) axis for reasoning that is native to, and intrinsically supported by, the architecture. We use experiments in Fig. 5 to demonstrate the test-time scaling potential of TTC-Net.

# D. Discussion

**More Memory-Based Architectures.** Beyond simple linear regression, it is also viable to perform parametric regression with a non-linear $M_t$ (e.g., letting $M_t$ be a neural network) (Sun et al., 2024; Zhang et al., 2025; Tandon et al., 2025), a loss functions other than the $\ell_2$ error (Behrouz et al., 2025c), a decaying or truncated $w_{t,\tau}$ (Gu et al., 2020; Behrouz et al., 2025a), different regularization terms (Von Oswald et al., 2023), more advanced optimization algorithms (Behrouz et al., 2024; 2025a; von Oswald et al., 2025; Peng et al., 2025b), such as Adam, Muon (Jordan et al., 2024), conjugate gradient, and Chebyshev iteration. Each such modification gives rise to a distinct memory-based architecture that is covered in Eq. (1). A comprehensive summary of these extensions can be found in Behrouz et al. (2025c). Another thread of work further combines multiple memory mechanisms, yielding hybrid sequence models (Ren et al., 2024; Dong et al., 2024; Zancato et al., 2024; Du et al., 2025).

**Difference from Amos et al. (2018).** Seminal work by Amos et al. (2018) proposed leveraging differentiable LQR as a trainable neural network layer for end-to-end control and planning for the first time. In contrast to Amos et al. (2018), we introduce a contextualized parameterization that adapts the underlying control problem dynamically to the input context. Moreover, while Amos et al. (2018) focuses on classical control tasks, our work applies an LQR-based module to enhance reasoning capabilities in LLMs. Finally, we propose a new scalable LQR solver that enables efficient computation of the LQR-based TTC layer, thereby making its integration into large-scale LLMs practical.

**Comparison with Test-Time Discovery.** TTD (Yuksekgonul et al., 2026) is a recent work that came out during the preparation of this paper. To the best of our knowledge, it is the only other approach that explores online RL–based adaptation at test time. However, the technical approaches differ substantially: TTD performs model-free, policy-based RL by updating the model parameters (i.e., slow weights), whereas our method adopts a model-based, value-based RL formulation that performs planning over the hidden states (i.e., fast weights) without modifying the model weights. We view these two approaches as complementary, and their integration may further enhance test-time planning and reasoning capabilities. We also make a table to illustrate the categories of test-time learning methods.

**Connection to World Models.** TTC generates the environment parameters $\{(\boldsymbol{A}_t, \boldsymbol{B}_t, \boldsymbol{Q}_t, \boldsymbol{R}_t)\}_{t=1}^{T}$ conditioned on the input context, and then solves a control problem defined over these parameters to produce the output. The resulting linear state transitions and quadratic cost together constitute a simplified world model (Ha & Schmidhuber, 2018; Hafner et al.,

| | Fast weight | Slow weight |
|---|---|---|
| SSL | DeltaNet(Yang et al., 2024b), GDN (Yang et al., 2024a) Titans(Behrouz et al., 2024), Nested Learning (Behrouz et al., 2025b) MesaNet (von Oswald et al., 2025), GatedKalmanNet(Peng et al., 2025b) | TTT-MLP (Sun et al., 2024) E2E-TTT (Tandon et al., 2025) |
| RL | TTC (Ours) | TTD (Yuksekgonul et al., 2026) |

*Table 3.* Taxonomy of various test-time learning methods.

2019; Hansen et al., 2022; 2023), capturing the essential dynamics and objectives relevant to the task. Rather than aiming for high-fidelity environment simulation, TTC emphasizes fast decision-making and learnability by solving this tractable world model efficiently and differentiating this system to train it in an end-to-end manner. From this perspective, TTC can be interpreted as an in-context world model: it infers a context-dependent and concise (solvable) representation of the "world", upon which it performs planning and decision-making. All parameters of this "world" are then directly trained through the supervision on the predicted actions.

**Connection to Continuous CoT.** Continuous Chain-of-Thought (CoT) (Hao et al., 2024) has been proposed to enable LLMs to reason beyond discrete tokens by representing intermediate reasoning traces as latent vectors. The general framework of continuous CoT repeatedly invokes a transformer backbone without decoding intermediate representations into tokens, while maintaining reasoning entirely within the latent space. Following this paradigm, an extensive body of work has focused on architectural innovations. For example, Geiping et al. (2025) injects perturbations and incorporates initial signals as inputs to all recurrent blocks. Wang et al. (2025a) introduces a two-level loop structure with additional supervision applied at each recurrent block. Other works further improve the scalability of looped transformers (Zeng et al., 2025; Zhu et al., 2025b). On the theoretical side, Giannou et al. (2023); Yang et al. (2023a); De Luca & Fountoulakis (2024); Saunshi et al. (2025) show that looped transformers can significantly enhance expressivity and may even achieve Turing completeness. In addition, Zhu et al. (2025a) demonstrates that latent CoT can encode the search frontier more efficiently. TTC-Net shares a similar high-level structure in that the "thinking" process is modeled within a latent state space. However, unlike continuous CoT methods, which are typically simulating an open-ended search, TTC-Net employs a closed-loop reasoning process, where rewards incurred at later steps are propagated backward to earlier stages to guide decision-making.

# E. More Experiments

## E.1. Ablation Studies

We conduct an ablation study on the MATH-500 benchmark to evaluate the effect of different design choices in TTC-Net. We examine three key design dimensions: (1) time modulation, (2) training horizon sampling strategy, and (3) TTC layer insertion interval. As we described in Sec. 4, the full model is a combination of three key features: (1) time-heterogeneous parameterization, (2) training horizon sampled from a Poisson log-normal distribution, and (3) an 8:1 ratio of attention to TTC layers. Each ablation study varies one of these three design dimensions while keeping the remaining components the same as in the full configuration. All model variants are trained with a fixed time horizon $T_{train} = 8$. At evaluation time, we assess generalization to different inference-time horizons $T_{test} \in \{8, 16\}$. We adopt the same recipe in Sec. 5.2 to train each model and report the test accuracy on MATH-500. All results are shown in Tab. 4.

**Time Homogeneity vs. Heterogeneity.** Time-heterogeneous parameterization allows the matrices $\{(\boldsymbol{A}_t, \boldsymbol{B}_t, \boldsymbol{Q}_t, \boldsymbol{R}_t)\}_{t=1}^T$ to vary across time steps $t$, whereas time-homogeneous parameterization adopts a shared set of parameters $(\boldsymbol{A}, \boldsymbol{B}, \boldsymbol{Q}, \boldsymbol{R})$ for all $t$. We implement the time-homogeneous variant by removing the time modulation coefficients. As shown in Tab. 4, the time-homogeneous TTC (rows 1-2) consistently underperforms its time-

*Table 4.* **Ablation studies.** Accuracy (%) on MATH-500 dataset.

| Variants | MATH-500 |
|---|---|
| Base model | 25.00 |
| **Parameterization** | |
| Homo. ($T_{test} = 8$) | 48.40 |
| Homo. ($T_{test} = 16$) | 45.70 |
| **Horizon sampling** | |
| Fixed ($T_{test} = 8$) | 50.60 |
| Fixed ($T_{test} = 16$) | 31.50 |
| Unif. ($T_{test} = 8$) | 50.80 |
| Unif. ($T_{test} = 16$) | 51.00 |
| **Interleaving ratio** | |
| Attn:TTC = 4:1 | 53.00 |
| Attn:TTC = 16:2 | 50.30 |
| Attn:TTC = 16:1 | 47.20 |
| **Full model** | |
| *Time Heter. + PLN + (8:1)* | |
| TTC-Net ($T_{test} = 8$) | 52.80 |
| TTC-Net ($T_{test} = 16$) | 53.60 |

heterogeneous counterpart (rows 10-11). Moreover, when the planning horizon is changed at test time, performance further degrades under the homogeneous setting. These results support our hypothesis that time-heterogeneous dynamics $\{(\boldsymbol{A}_t, \boldsymbol{B}_t, \boldsymbol{Q}_t, \boldsymbol{R}_t)\}_{t=1}^{T}$ are critical for modeling complex latent-space dynamics. In contrast, time-uniform linear dynamics suffer from limited expressivity. Additionally, without the discounting effect of the time modulation, generalization across planning horizons deteriorates.

**Horizon Sampling Strategy.** We further consider two alternative training horizon sampling strategies besides the Poisson log-normal (PLN) distribution: (1) a fixed horizon with $T_{train} = 8$, and (2) a uniform distribution with $T_{train} \sim \mathrm{Unif}(1, 32)$. According to Tab. 4, training with a fixed horizon yields comparable performance at the matched test horizon (row 3), but fails to generalize or further improve when evaluated with a larger test-time horizon (row 4). The uniform horizon sampling strategy (rows 5–6) achieves accuracy on par with the full model (rows 10–11). However, this comes at a substantially higher training cost. Larger planning horizons require increased computational time. The sampled horizon from a PLN distribution concentrates around the mean value of 8, whereas the uniform distribution nearly doubles the average training horizon.

**Interleaving Pattern between Attention and TTC.** We further study the interleaving ratio between attention and TTC layers. We use Attn:TTC = $n : m$ to denote that $m$ TTC layers are inserted after every $n$ attention layers. Our default configuration is an $8 : 1$ ratio in a 32-layer Transformer. The results are reported in rows 7–9 of Tab. 4 We observe that increasing the number of TTC layers can monotonically improve overall accuracy (rows 7 and 9). However, this improvement comes at a higher computational cost and is less cost-effective than adopting a relatively smaller interleaving ratio while increasing the test-time horizon (rows 7 and 11). Moreover, comparing configurations that stack more TTC layers consecutively (row 8) with those that interleave them more uniformly with attention layers (row 10), the results ($8 : 1$ vs $16 : 2$) indicate that distributing TTC layers evenly throughout the network yields better performance. These observations justify our default choice of $8 : 1$.

### E.2. Comparison with Decoding-Based Scaling

A common approach to improving reasoning at test time is to allocate additional decoding computation. Methods such as Best-of-$N$ sampling (BoN) (Stiennon et al., 2020), self-consistency (Wang et al., 2022), Tree-of-Thoughts (ToT) (Yao et al., 2024), and planning-based decoding methods such as RAP (Hao et al., 2023) improve performance by sampling, filtering, or searching over candidate reasoning traces. Unlike TTC-Net, which incorporates planning as an internal architectural mechanism over latent space, these approaches instantiate planning externally: the base model remains unchanged, while additional inference-time procedures are applied on top of autoregressive generation.

To compare these two forms of test-time computation, we evaluate TTC-Net against BoN and ToT under the same reasoning setting. All methods use chain-of-thought prompting. Tab. 5 shows that TTC-Net achieves higher accuracy while generating substantially fewer tokens. The gap in output length is particularly significant: external search methods require more than 20K generated tokens on average, whereas TTC-Net produces more concise reasoning traces. This suggests that internal latent planning can improve reasoning quality without relying on extensive enumeration and filtering in the token space. Nevertheless, TTC is complementary to decoding-based scaling techniques, and they can be combined

*Table 5.* Comparison with decoding-based test-time scaling methods.s TTC-Net improves accuracy while requiring substantially fewer generated tokens.

| Method | Acc. | Avg. Tokens |
|---|---|---|
| BoN | 22.43 | 20731.10 |
| ToT | 21.08 | 22412.02 |
| TTC-Net | **23.34** | **2801.50** |

to achieve stronger performance. TTC differs in that the planning computation is amortized through the model architecture.

We also isolate the interaction between TTC and explicit chain-of-thought generation. To disable explicit CoT for LLMs, we enforce the following system prompt:

```
1  Solve math problems silently. Do not disclose chain-of-thought, scratch work, intermediate
       reasoning, or hidden deliberation.
2
3  Return only the final answer in this exact format:
4
5  \[
6  \boxed{...}
7  \]}
```

Tab. 6 reports a $2 \times 2$ comparison on AMC, varying whether the model includes TTC layers and whether the inference prompt encourages explicit CoT. TTC improves performance in both regimes. Without explicit CoT, TTC increases accuracy from $3.92\%$ to $7.83\%$, indicating that latent planning provides a meaningful internal reasoning even when token-space reasoning steps are suppressed. With CoT enabled, TTC further improves accuracy from $20.78\%$ to $23.34\%$, showing that TTC and explicit reasoning traces are complementary.

*Table 6.* Ablation study between TTC and explicit CoT prompting on AMC.

| Model | w/ CoT | w/o CoT |
|---|---|---|
| w/o TTC | 20.78 | 3.92 |
| w/ TTC | **23.34** | **7.83** |

We note that TTC should not be viewed as a replacement for chain-of-thought reasoning. Instead, TTC improves the latent computation underlying each generated reasoning step. Explicit CoT remains important for difficult problems, while TTC provides an additional internal planning mechanism that can reduce reliance on long or redundant decoding-time search.

### E.3. Evaluation beyond Reasoning Benchmarks

Although our primary goal is to improve reasoning through architectural design, we also evaluate whether TTC-Net preserves general language modeling and understanding capabilities beyond math reasoning benchmarks. We evaluate the fine-tuned LLaMA-based checkpoints on standard LM evaluation benchmarks, including commonsense reasoning, reading comprehension, knowledge-intensive multiple-choice evaluation, and code generation. The results are reported in Tab. 7.

*Table 7.* **Evaluation beyond math reasoning benchmarks.** TTC-Net improves several reasoning-heavy and code-related tasks while maintaining competitive performance on general language understanding benchmarks.

| Model | HellaSwag | WinoGrande | ARC-C | MMLU | HumanEval |
|---|---|---|---|---|---|
| LLaMA + SFT | **54.01** | **74.74** | 51.79 | 59.31 | 32.32 |
| TTC-Net | 53.75 | 72.09 | **59.22** | **64.02** | **49.39** |

TTC-Net obtains gains on ARC-Challenge, MMLU, and HumanEval, which require long-horizon reasoning or program synthesis ability. On HellaSwag and WinoGrande, TTC-Net remains competitive with the raw SFT baseline, suggesting that the additional planning module preserves general language modeling ability learned in base models. These results support the generality of TTC beyond the math benchmarks used in our main evaluation.

### E.4. Evaluation on Qwen Models

We further validate TTC-Net on a stronger reasoning backbone, Qwen2.5-Math-7B. TTC layers are modular and can be inserted into transformer-based architectures without relying on a specific backbone. As shown in Tab. 8, TTC-Net improves over standard full-parameter fine-tuning across all evaluated math benchmarks, confirming the consistency of our proposed methods.

*Table 8.* **Results on Qwen2.5-Math-7B.** TTC-Net consistently improves over standard fine-tuning on a stronger reasoning backbone.

| Model | MATH-500 | AMC | AIME24 | AIME25 |
|---|---|---|---|---|
| Qwen2.5-Math-7B | 43.8 | 33.0 | 6.7 | 6.7 |
| + Fine-tuning | 74.4 | 51.7 | 20.4 | 18.7 |
| + TTC-Net | **75.6** | **55.4** | **22.9** | **20.8** |

### E.5. Computational Analysis

TTC-Net is designed to preserve efficient inference with hardware-level optimization. A naive implementation of finite-horizon optimal control would introduce substantial overhead, especially when the planning horizon is large. TTC-Net avoids this bottleneck through a hardware-friendly LQR solver and fused CUDA kernels that reduce memory I/O traffic.

*Table 9.* Inference throughput of Transformer and TTC-Net on an NVIDIA H100 GPU.

| Model | Tokens/s |
|---|---|
| Transformer | 47.16 |
| TTC-Net ($T = 8$) | 45.77 |
| TTC-Net ($T = 16$) | 44.61 |
| TTC-Net ($T = 64$) | 41.43 |
| TTC-Net ($T = 128$) | 36.18 |

Tab. 9 reports inference throughput on an NVIDIA H100 GPU. Compared with a standard transformer, TTC-Net introduces a small overhead at short planning horizons. At $T = 8$, throughput decreases from $47.16$ to $45.77$ tokens/s. Increasing the horizon leads to a gradual reduction in throughput, but the model remains practical even at $T = 128$. This behavior reflects the intended accuracy-efficiency trade-off: larger horizons allocate more computation to latent planning, while the parallel solver and fused implementation keep the additional cost moderate.

## E.6. Analysis of Latent Trajectories

The central hypothesis behind TTC-Net is that the explicit control mechanism can internalize meaningful planning trajectories. In this section, we testify this hypothesis by analyzing TTC latent states and measuring how they align with the explicit reasoning process specific to targeted tasks.

**Linear Probing on Sudoku.** We study whether TTC latent states become progressively more informative over the planning horizon. On Sudoku, we train a linear probe to decode intermediate TTC latent states into per-cell digit predictions. The TTC states themselves are not directly supervised with cell-level labels during model training; only the final outcome is used to make the final prediction. Tab. 10 shows that cell-level decoding accuracy increases with the planning step, from $42.50\%$ at step 0 to $50.38\%$ at step 8. This trend indicates that TTC planning progressively refines the latent representation toward the terminal solution.

*Table 10.* Linear probing accuracy of TTC latent states on Sudoku.

| Step | Cell Acc. |
|------|-----------|
| 0 | 42.50 |
| 2 | 42.96 |
| 4 | 46.20 |
| 8 | **50.38** |

**Latent Trajectory Alignment with CoT.** We measure the alignment between TTC latent trajectories and CoT token representations. Let $\boldsymbol{X} \in \mathbb{R}^{L \times d}$ denote token representations extracted from explicit CoT traces at the corresponding residual stream layer, where $L$ is the CoT length. Likewise, we let $\boldsymbol{H} \in \mathbb{R}^{T \times d}$ denote the latent trajectory produced by TTC over a planning horizon of length $T$. Note that we extract $\boldsymbol{H}$ from the first token after the prompt. Because TTC planning steps and CoT tokens need not align one-to-one, we compute a monotone-path alignment score:

*Table 11.* Trajectory-level alignment between TTC latent dynamics and explicit CoT representations.

| Pairing | Similarity |
|---------|-----------|
| Same question | **0.534** |
| Random pairs | 0.212 |

$$\text{sim}(\boldsymbol{H}, \boldsymbol{X}) = \max_{p \in \mathcal{P}} \frac{1}{|p|} \sum_{(i,j) \in p} \frac{\boldsymbol{H}_i^\top \boldsymbol{X}_j}{\|\boldsymbol{H}_i\|_2 \|\boldsymbol{X}_j\|_2}, \tag{39}$$

where $\mathcal{P} = \{((i_i, j_1), \ldots, (j_m, j_m)) | (i_1, j_1) = (1, 1), (i_m, j_m) = (T, L), (i_{k+1} - i_k, j_{k+1} - j_k) \in \{(1, 0), (0, 1), (1, 1)\}\}$ denotes the set of monotone alignment paths between TTC planning steps and CoT token positions. The similarity score above intends to match each TTC latent to a consecutive chunk of tokens' representations, and the maximization problem can be solved via dynamic programming. We compare this similarity score between latent trajectories and CoT representations derived from the same question and different question pairs. Tab. 11 reports the resulting similarity on AMC. TTC latent trajectories are more aligned with CoT representations from the same question than with randomly paired trajectories. This suggests that the TTC dynamics are highly correlated with the explicit reasoning trace, internalizing token-space CoT with a more compact latent space.

## E.7. Experiment Details

**Sudoku Solving.** All experiments use a 32-layer model with 4 attention heads and an embedding dimension of 128, trained using a batch size of 16 and a learning rate of $5 \times 10^{-3}$ with 10% tokens for warm-up and decay to $5 \times 10^{-4}$. The proposed TTC layer operates with a hidden and control dimension $d = 16$, employs 4 parallel TTC heads, and uses $r = 16$ for basis matrices $\{\boldsymbol{Q}^{(i)}\}_{i=1}^r$ and $\{\boldsymbol{B}^{(i)}\}_{i=1}^r$. Models are trained and evaluated on standard reasoning datasets, with 9,000 training samples and 1,000 test samples, following consistent data splits across experiments. We further provide a pseudo-code in Algorithm 6 to illustrate the multi-step completion process of a Sudoku game.

**Math Reasoning.** For math reasoning tasks, all models are fine-tuned using AdamW with a global batch size of 96, learning rate of $2 \times 10^{-5}$, weight decay of 0.1, and a cosine learning rate scheduler using 10% tokens for warm-up and then decaying to $2 \times 10^{-6}$. Each TTC layer has 16 heads, with each head operating on a state space of dimension 16 ($d = 16$). We set the number of basis to $r = 16$ for both $\{\boldsymbol{Q}^{(i)}\}_{i=1}^r$ and $\{\boldsymbol{B}^{(i)}\}_{i=1}^r$. TTC training employs stochastic planning horizons sampled from a Poisson–lognormal distribution, with a mean horizon of 8 and support ranging from 1 to 32 steps. During evaluation, we sample 8 solutions for each problem in AIME and AMC, using a temperature of 0.6. For Math-500, we employ greedy decoding to generate a single solution per problem.

**Reasoning Data Curation.** The curated data collection we used for fine-tuning models comprises questions from recent (2024–2025) verifiable reasoning datasets spanning mathematics, multi-domain academic subjects, science, medicine,

---

**Algorithm 6** Multi-step Sudoku completion

---

**Require:** An incomplete board $B_{init} \in \{[\texttt{mask}], 1, \cdots, 9\}^{9 \times 9}$.
**Ensure:** Completed board $B \in \{1, \cdots, 9\}^{9 \times 9}$.
 1: Initialize board $B \leftarrow B_{init}$.
 2: **while** $\exists (i,j)$ such that $B_{ij} = [\texttt{mask}]$ **do**
 3:     Initialize prediction $P \in \{1, \cdots, 9\}^{9 \times 9}$, confidence $C \in [0,1]^{9 \times 9} \leftarrow 0$.
 4:     Obtain per-cell digit probabilities: $X \leftarrow \text{Predictor}(B) \in (0,1)^{9 \times 9 \times 9}$.       ▷ Invoke neural network backbone
 5:     **for** each masked cell $(i,j)$ such that $B_{ij} = [\texttt{mask}]$ **do**
 6:         $P_{ij} \leftarrow \arg\max_k X_{ijk}$       ▷ Most likely digit for cell $(i,j)$
 7:         $C_{ij} \leftarrow \max_k X_{ijk}$       ▷ Confidence of the prediction
 8:     **end for**
 9:     Select most confident masked cell: $(i^*, j^*) \leftarrow \arg\max_{(i,j)} C_{ij}$.
10:     Fill selected cell with its prediction: $B_{i^*,j^*} \leftarrow P_{i^*,j^*}$.
11: **end while**
12: **Return** $B$.

---

finance, and coding, totaling on the order of 4 million objectively checkable QA instances. One of the primary motivations for selecting more recent datasets is to mitigate the spurious performance caused by potential data leakage and contamination, a common concern in large-scale industrial pre-trained models (Shojaee et al., 2025). A substantial portion is math-centric (e.g., DeepMath-103K, Big-Math, Massive-Math-455K, ORZ-72k, DeepScaleR), emphasizing problems with unique numeric or symbolic answers suitable for rule-based verification and reinforcement learning. Broader datasets (e.g., Multi-Subject-RLVR, II-Thought, Natural Reasoning, General-Reasoner , Nemotron-CrossThink) extend coverage to physics, social sciences, law, and engineering, often filtering for short, automatically verifiable answers. Domain-specific subsets target high-stakes fields such as medicine (MedReason, Medical-O1, Medical-O1-SFT) and finance (e.g., Fino1). Building on these QA pairs, we further generate chain-of-thoughts (CoT) using `gpt-oss` for those whose intermediate reasoning traces are not given. To ensure correctness, we retain only those CoTs that pass 32 independent rounds of verifications by `gpt-oss`. After deduplication, we randomly sample and keep 800K examples as the final curated SFT dataset.

**Efficiency Benchmark.** In Fig. 3, we evaluate computational efficiency on an NVIDIA H200 GPU. Both throughput and memory footprint are measured over a complete forward and backward pass. Throughput is measured in GB/s, reporting the median runtime together with the 20-80% percentile statistics to account for runtime variability. We benchmark two scaling dimensions: (1) planning horizon $T \in \{2^4, \ldots, 2^{11}\}$ with logarithmic spacing, and (2) batch size $B \in \{2^4, \ldots, 2^{13}\}$. Across all results, we set the state dimension $d = 16$. Throughput is computed based on the theoretical FLOPs $BTd^3$ of solving $B$-many batched $d$-dimensional LQR problem with horizon $T$ via Riccati iteration and normalized by wall-clock latency. Memory footprint is evaluated separately by measuring peak GPU memory usage during execution, reported in GB. For memory benchmarking, we fix either $B = 1024$ (when varying $T$) or $T = 64$ (when varying $B$) to isolate scaling effects. All benchmarks compare different implementation providers under identical input generation and precision settings, and out-of-memory cases are recorded as zero throughput.

