# OpenReview forum: "Beyond Test-Time Memory: State-Space Optimal Control for LLM Reasoning"
_ICML.cc/2026/Conference — ICML 2026 regular_

### Official Review · Reviewer_ghUw · 2026-03-10

**Soundness:** 3
**Presentation:** 3
**Significance:** 3
**Originality:** 3
**Overall Recommendation:** 4
**Confidence:** 4

**Summary:**

This paper models reasoning as an explicit optimal control problem and proposes TTC-Net, a unified architecture that integrates test-time control layers to enable planning before prediction. The theoretical contributions are sound, the hardware-efficient LQR solver is a significant engineering contribution, and the empirical results on Sudoku and math reasoning are compelling.

**Compliance With Llm Reviewing Policy:**

Affirmed.

**Ethical Review Concerns:**

The experimental results resolved my concerns, and I maintain my score.

**Final Justification:**

Given the author responses, I tend to maintain my score.

**Key Questions For Authors:**

1. Does TTC replace the need for CoT or other test-time strategies, or are they orthogonal?
2. How crucial is the full LQR formulation? Could a simpler, non-iterative planning step achieve similar results?
3. How much does the diagonal structure of $A_t$ and $R_t$ affect final performance compared to a dense version (if it were computationally feasible)?

**Limitations:**

Yes

**Strengths And Weaknesses:**

Strengths:
1. This paper introduces a new architectural paradigm, which formulates reasoning as optimal control and introduces the TTC layer. This bridges the gap between memory retrieval and reinforcement learning.
2. The derivation of the symplectic iteration (Theorem 3.3) is a key theoretical contribution, transforming a sequential compute-bound process into a parallelizable one that is more suitable for modern hardware.
3. This paper dives deep into the system-level implementation, performing kernel fusion, structured parameterization, numerical stability, and caching for the backward pass. The empirical validation of Figure 3 shows the efficiency of hardware/software co-design.

Weaknesses:
1. The paper reports accuracy gains on Sudoku and MATH, but it does not report the corresponding inference latency or training throughput (tokens/second) for the TTC-Net compared to the baselines (Transformer, Mamba, etc.). We know TTC-Net is more accurate, but we don't know if it is 2x slower or 20% slower. This makes it difficult to assess the practical trade-off.
2. The paper compares with other architectural adapters (Mamba, Samba). However, a major recent trend is inference-time strategies like Chain-of-Thought, self-consistency, or tree-of-thoughts. A comparison or discussion of how TTC layers relate to or could complement these methods would strengthen the paper.
3. The paper would benefit from more extensive ablation studies to isolate the source of the gains. For instance, the author states that diagonalized SSMs can be as expressive as dense SSMs, yet lacks a performance comparison.

---

> ### Author Rebuttal · Authors · 2026-03-31
>
> We thank Reviewer ghUw for recognizing our contributions to bridge memory, reinforcement learning, and hardware-friendly control solvers. Please find our point-by-point responses below.
>
> **1. What is the inference latency and throughput trade-off compared to baselines?**
>
> We appreciate this suggestion from Reviewer ghUw. We benchmark the runtime of a standard transformer and TTC-Net (i.e., transformers augmented with TTC layers). Throughput is measured on an NVIDIA H100 GPU and reported in the following table. TTC introduces minimal computational overhead due to hardware-efficient LQR solvers and kernel fusion. Compared to raw transformer (after full fine-tuning), TTC-Net improves reasoning performance by 6% with slightly lower inference throughput (down 1.38 tokens/s) when horizon is 8. This highlights a favorable trade-off between accuracy and efficiency.
>
> |Model|Throughput (tokens/s)|
> |---|---|
> |Transformer|47.16|
> |TTC-Net ($T=8$)|45.77|
>
> **2. How does TTC relate to or complement inference-time strategies such as CoT, self-consistency, or tree-of-thoughts?**
>
> Our primary contribution lies at the architectural level, thereby models such as Mamba and Samba become the most relevant baselines. Note that all models in our paper, including TTC, use CoT prompting by default. TTC does not replace CoT. Instead, it enhances each step of CoT generation by incorporating planning, enabling more accurate and consistent next-token predictions. Methods such as self-consistency, ToT, and RAP operate as external search or sampling procedures at decoding time. In contrast, TTC implements an internal planning mechanism within the model, learning an implicit RL policy in latent space. TTC is orthogonal to these approaches and can be combined with them. External search (e.g., ToT [1], RAP [2]) can still be applied on top of TTC-enhanced models.
>
> We further conduct a *direct comparison between internal RL architecture (TTC-Net) and external RL-based planning approaches (BoN, ToT)*. Specifically, we evaluate the reasoning accuracy of TTC-Net against BoN and ToT. The results are presented in the table below.
>
> |Method|Accuracy|Avg Token|
> |---|---|---|
> |BoN|22.43|20731.10|
> |ToT|21.08|22412.02|
> |TTC-Net (Ours)|23.34|2801.50|
>
> Our findings demonstrates **TTC outperforms BoN and ToT in reasoning accuracy**. In the meanwhile, **TTC generates shorter and more straightforward reasoning chain, achieving more token-efficient reasoning**. This further supports the benefit of internal planning over external sampling-based search.
>
> **3. Ablation studies**
>
> Thanks for suggesting ablations on key design choices. We study three components of TTC-Net: **(1) Time-homogeneous vs. time-heterogeneous parameterization, (2) TTC layer insertion interval, (3) Training horizon sampling strategy**. The results are reported in the following table.
>
> We observe that **time-homogeneous parameterization leads to lower performance and poor horizon extrapolation**, highlighting the importance of temporal flexibility. Similarly, our horizon sampling strategy is critical: **uniform/random sampling enables robust extrapolation, Poisson log-normal sampling maintains extrapolation while reducing average training cost**. Insertion interval affects the balance between compute and performance, with moderate spacing yielding the best trade-off. These results validate our design choices and will be included in the final version.
>
> |Variants|MATH-500 (%)|
> |---|---|
> |Base model|25.00|
> |**Parameterization**||
> |Homo. ($T_{test}=8$)|48.40|
> |Homo. ($T_{test}=16$)|45.70|
> |**Horizon sampling**||
> |Fixed ($T_{test}=8$)|50.60|
> |Fixed ($T_{test}=16$)|31.50|
> |Unif. ($T_{test}=8$)|50.80|
> |Unif. ($T_{test}=16$)|51.00|
> |**Interleaving ratio**||
> |Attn:TTC = 4:1|53.00|
> |Attn:TTC = 16:2|50.30|
> |Attn:TTC = 16:1|47.20|
> |**Full model**||
> |*Time Heter. + PLN + (8:1)*||
> |TTC-Net ($T_{test}=8$)|52.80|
> |TTC-Net ($T_{test}=16$)|53.60|
>
> We further note that *fully dense parameterization of transition matrices is non-trivial to scale without further algorithm optimization*. Richer structured parameterizations (e.g., low-rank + diagonal [1]) may offer a promising direction.
>
> [1] Yang et al., Parallelizing Linear Transformers with the Delta Rule over Sequence Length
>
> **4. Is the full LQR formulation necessary, or can simpler planning suffice?**
>
> LQR represents one of the simplest yet principled formulations of planning, based on linear approximation of system dynamics and quadratic approximation of cost functions. In fact, many advanced nonlinear control methods (e.g., interior-point methods (IPM), sequential quadratic programming (SQP)) iteratively solve LQR subproblems, highlighting its foundational role. LQR provides a tractable and stable optimization objective, enabling efficient integration into neural architectures. Therefore, LQR should be viewed as a minimal yet sufficient abstraction that captures essential planning structure and remains computationally feasibility and efficiency.

---

> > ### Author Rebuttal · Reviewer_ghUw · 2026-04-03
> >
> > The experimental results resolved my concerns, and I maintain my score.

---

### Official Review · Reviewer_AME7 · 2026-03-11

**Soundness:** 2
**Presentation:** 2
**Significance:** 3
**Originality:** 3
**Overall Recommendation:** 4
**Confidence:** 1

**Summary:**

This paper proposes TTC-Net, a hybrid architecture that inserts a Test-Time Control layer into a standard backbone, aiming to introduce explicit planning into sequence modeling. The idea of reframing reasoning as control rather than pure memory/retrieval is interesting, and the empirical results on Sudoku and math benchmarks are promising.

**Compliance With Llm Reviewing Policy:**

Affirmed.

**Final Justification:**

In light of the authors' response,  I tend to maintain my scores.

**Key Questions For Authors:**

See weaknesses.

**Limitations:**

yes

**Strengths And Weaknesses:**

## Strengths
The paper is conceptually novel and ambitious. I like the core intuition that reasoning may benefit from an explicit planning inductive bias built into the architecture, rather than relying only on scaling or post-training. The results are encouraging enough to suggest this is a worthwhile direction.

## Weaknesses
- Experimental setting is somewhat odd. For math reasoning, the chosen backbone is not the strongest or most standard one. Using a stronger reasoning backbone, such as Qwen, would make the empirical case more convincing.

- The method is hard to follow. The motivation is clear, but the technical section is difficult to understand. For instance, the terminology KKT and the derivation of Equation 1 are not presented clearly enough.

- The conceptual claim is broader than the implementation. The paper argues for planning-centric reasoning, but the actual method is a specific linear-quadratic control instantiation. It would help to better justify why this is the right abstraction.

I am borderline accept. I find the idea original and potentially impactful, but the paper is harder to follow than it should be, and the empirical setting is not yet fully convincing. My confidence is low (since I am not an expert in model architecture or linear attention), and I would especially consider the opinions of reviewers with stronger expertise in this architectural/control-theoretic area.

---

> ### Author Rebuttal · Authors · 2026-03-31
>
> We thank Reviewer AME7 for highlighting our paper is novel and ambitious, as well as the constructive comments to improve our work. We have provided additional experiments and futher clarification to address your concerns. Please see our rebuttal below.
>
> **1. Stronger reasoning backbone (e.g., Qwen) for evaluation?**
>
> We thank the reviewer for this suggestion. We initially chose the LLaMA family as our primary backbone because it is less susceptible to data contamination, allowing for a cleaner evaluation of reasoning improvements before and after fine-tuning. Importantly, the TTC layer is fully modular and backbone-agnostic, and can be integrated into any transformer-based architecture.
>
> To validate this, we construct TTC-Net on top of Qwen-2.5-Math-7B and fine-tune it under the same dataset and recipe described in Sec. 5.2. We compare against standard full-parameter fine-tuning. The results shown below indicates that TTC consistently improves reasoning performance on a stronger backbone, like Qwen.
>
>
> | Model | Math500 | AMC | AIME24 | AIME25 |
> |-|-|-|-|-|
> | Qwen2.5-Math-7B | 43.8 | 33.0 | 6.7 | 6.7 |
> | + Fine-tuning | 74.4 | 51.7 | 20.4 | 18.7 |
> | + TTC-Net | 75.6 | 55.4 | 22.9 | 20.8 |
>
> We will continue to run and include more baselines listed in Tab. 2.
>
> **2. Writing clarity.**
>
> Karush-Kuhn-Tucker (KKT) conditions (Sec. 3.2) provide necessary optimality conditions for constrained optimization problems via Lagrangian methods, as described in [1]. In our work, the KKT conditions are introduced to formalize the necessary definitions and notation used in deriving our result on differentiation through the optimal solution (Theorem 3.2).
> Equation (1) is not our contribution, but rather a formulation adopted from prior work showing that sequential architectures can be interpreted as optimization-driven memory mechanism. A friendly introductory materials can be found in [2].
> We will clarify these explicitly to avoid confusion. Detailed derivations to obtain all of our results have been included in the current manuscript.  To further improve accessibility and reproducibility, we will add pseudocode for TTC forward/backward passes. We welcome further suggestions from the reviewers to improve clarity.
>
>
> [1] Boyd and Vandenberghe, Convex Optimization, Chapter 5.5.3
>
> [2] Behrous et al., Titans: Learning to memorize at test time
>
> **3. Why is linear-quadratic control the appropriate abstraction for planning-centric reasoning?**
>
> LQR is a principled and widely adopted abstraction for planning due to its tractability and approximation power.
>
> 1. LQR can be interpreted as a first-order approximation of nonlinear dynamics combined with a quadratic approximation of the reward (cost) function. This makes it a natural local surrogate (convex relaxation) for more complex planning problems.
> 2. The resulting optimization problem is tractable, allowing analytic and stable solutions. When viewing LQR as a convex upper bound of a non-linear complicated control problem, LQR plays a role analogous to gradient descent, where a simple local model provides a tractable update rule for a more complex objective.
> 3. In our setting, LQR operates in a high-dimensional latent space of LLMs, where learnable linear dynamics can be expressive enough in approximating non-linear dynamical systems (as seen in [1]). Moreover, when incorporated into LLMs, TTC layers are stacked, complementing expressivity beyond a single LQR system.
>
> Taken together, LQR provides a balanced trade-off between expressivity and tractability, serving as a suitable abstraction for integrating planning into scalable sequence models.
>
> [1] Gu et al., Combining Recurrent, Convolutional, and Continuous-time
> Models with Linear State-Space Layers

---

> > ### Author Rebuttal · Reviewer_AME7 · 2026-04-01
> >
> > Thanks for the authors' reply, and I maintain my score.

---

### Official Review · Reviewer_Ccnu · 2026-03-13

**Soundness:** 3
**Presentation:** 3
**Significance:** 3
**Originality:** 3
**Overall Recommendation:** 4
**Confidence:** 3

**Summary:**

The authors propose Test-Time Control (TTC), a way to model reasoning as an optimal
control problem over internal representations within a sequential model architecture.
They introduce the TTC layer which synthesizes an MDP with linear state transitions
and quadratic cost functions, and use receding horizon control to select the first-step
optimal action corresponding to the optimal next-token representation. They propose a
new symplectic LQR solver and claim that it can be efficiently integrated with existing
hardware. The authors compare with baselines on reasoning tasks such as Sudoku and
show that their method can consistently achieve the highest levels of performance.

**Compliance With Llm Reviewing Policy:**

Affirmed.

**Key Questions For Authors:**

● Is there a way to visualize the computed latent state trajectories, using something like a t-SNE plot on a reasoning task such as Sudoku? Having such a visualization would be good for understanding the paper and whether the TTC reasoning is doing the right thing.

● Do the authors notice any pathological cases arising in experimentation where using linear dynamics leads to reasoning flaws? This question is tied to the previous visualization question.

● How does a CoT baseline compare with your method for your experiments?

● Have you tried longer horizons (e.g., $T >= 128$) for training? Do you notice any numerical instability and/or sensitivity in the matrices defining your MDP?

● Can you compare time-per-inference between TTC and standard transformers / other
baselines? How does this vary with horizon $T$?

**Limitations:**

The authors mention that future work could involve improving any/all parts of training, and a
more comprehensive evaluation of their method on larger-scale models.

**Strengths And Weaknesses:**

Strengths:

● The paper’s goals are clear and the proposed method is interesting.

● TTC can effectively do longer-horizon reasoning similar to chain-of-thought (CoT) but without increasing output length. More specifically, they can increase some planning horizon $T$ to do reasoning via solving an MDP under the hood.

● The authors impressively resolved their I/O bottleneck during the inference optimization by using fused CUDA kernels, which enables a control-theoretic approach such as that described in this paper.

● The experimental results compared to baselines appear good, although error bars are
not reported so statistical significance cannot be evaluated.

Weaknesses:

● The authors assume linear dynamics and use LQR (which assumes convex cost landscape) in the TTC layer MDP, which may not be appropriate for solving reasoning tasks that may have, e.g., discrete conditional logic that may not be easily represented in this way.

● There is not enough analysis or visualization on whether the latent trajectories make sense.

● The authors should include CoT baseline to compare “external” planning with their
proposed “internal” planning scheme.

● The internal MDP is synthesized online at test-time, and there is no analysis on the sensitivity of, e.g., the basis matrices $Q$ and $B$, to different tasks. For example, if the training horizon was increased from a mean of 8 (as noted in the paper) and max of 32, to something higher such as $T > 128$, does the model remain numerically stable?

● The proposed method is quite complex, with many mathematical tricks and hardware dependencies (e.g., CUDA) to make implementing the method practical.

---

> ### Author Rebuttal · Authors · 2026-03-31
>
> We thank Reviewer Ccnu for noting the interesting motivation and impressive implementation of our work. Please see our detailed responses below.
>
> **1. Does the LQR assumption limit reasoning capability?**
>
> While classical LQR assumes linear dynamics and quadratic costs, this does not fundamentally limit expressivity:
>
> * **Latent-space expressivity.** High-dimensional continuous representations are strictly more expressive than discrete symbolic states, as they can encode superpositions of multiple reasoning trajectories or search frontiers simultaneously, rather than committing to a single discrete path [1].
> * **Implicit modeling of discrete reasoning.** Dense linear operators in latent space can approximate sparse, structured transitions corresponding to logical operations. Hence, discrete reasoning patterns can be embedded and recovered from dense parameters.
> * **Depth-induced expressivity.** Although a single TTC layer relies on simplified assumptions (e.g., convexity), stacking multiple TTC layers yields a highly expressive architecture, analogous to standard deep networks where simple components compose into complex functions.
>
> Therefore, the LQR formulation does not limit reasoning capability, but instead provides a tractable yet expressive modeling of planning.
>
> [1] Zhu et al., Reasoning by Superposition: A Theoretical Perspective on Chain of Continuous Thought.
>
> **2. Interpretation and analysis of latent trajectories**
>
> We thank the reviewer for this suggestion. To analyze interpretability, we propose a trajectory-level alignment metric between TTC latent dynamics and explicit chain-of-thought (CoT) reasoning. Let $X \in \mathbb{R}^{L \times d}$ be token representations from CoT traces (extracted from the residual stream at the corresponding layer where TTC layers are inserted), and $H \in \mathbb{R}^{T \times d}$ be latent trajectories generated by TTC. We define a trajectory similarity score: $\max_{p \in \mathcal{P}} \frac{1}{|p|} \sum_{(i,j) \in p} \cos(H_i, X_j)$, where $\mathcal{P}$ denotes monotone paths.
>
> We compute this similarity between latent dynamics and CoT token features for each question in AMC, and compare it to randomly paired (uncorrelated) trajectories as a reference. The results are reported below:
>
> |Same Questions|Random Pairs|
> |-|-|
> |0.534|0.212|
>
> We observe that latent trajectories exhibit higher alignment with CoT representations on the same question compared to mismatched ones. This indicates that TTC dynamics internalize the explicit reasoning process rather than encoding uncorrelated features. Importantly, TTC does not aim for one-to-one alignment, as its latent dynamics can encode multiple reasoning frontiers simultaneously, leading to softer alignment.
>
> **3. Compare TTC with CoT baselines**
>
> We clarify that TTC is complementary to CoT, not a replacement. All methods (including ours and baselines) operate under CoT prompting. CoT alone does not inherently include external planning. In contrast, BoN, ToT [1], and RAP [2] correspond to external planning approaches via rejection sampling or tree search at inference time. We compare TTC-Net with Llama-3-8B-Instruct under these decoding strategies:
>
> |Method|Accuracy|Avg Token|
> |---|---|---|
> |BoN|22.43|20731.10|
> |ToT|21.08|22412.02|
> |TTC-Net|23.34|2801.50|
>
> TTC achieves consistently stronger reasoning performance than BoN and ToT, while producing significantly shorter outputs. This indicates more efficient reasoning and suggests that internal planning is more token-efficient than external search methods.
>
> [1] Yao et al., Tree of Thoughts: Deliberate Problem Solving with Large Language Models
> [2] Hao et al., Reasoning with Language Model is Planning with World Model
>
> **4. Sensitivity to horizon and parameters?**
>
> We design TTC with robustness to long horizons and numerical stability in mind. When solving LQR, we employ symplectic iterations with a stepwise normalization technique (Appendix C), preventing numerical overflow. In our reward parameterization, we introduce decaying rewards along the horizon, improving conditioning and stability.
>
> During training, instead of fixing a long horizon, we use horizon sampling, enabling out-of-the-box extrapolation to longer horizons at inference. Empirically, TTC-Net supports horizons up to 128 steps without numerical instability. We did not adopt $T=128$ during training because *moderate horizons already generalize well, improving efficiency*.
>
> **5. Efficiency comparison**
>
> We benchmark inference throughput of standard transformers and TTC-Net on an NVIDIA H100 GPU. TTC introduces minimal overhead due to hardware-efficient LQR solvers and kernel fusion. Runtime scales primarily with the planning horizon but remains moderate:
>
> |Model|Throughput (tokens/s)|
> |---|---|
> |Transformer|47.16|
> |TTC-Net ($T=8$)|45.77|
> |TTC-Net ($T=16$)|44.61|
> |TTC-Net ($T=64$)|41.43|
> |TTC-Net ($T=128$)|36.18|
>
> This demonstrates that TTC maintains competitive efficiency while enabling internal planning.

---

> > ### Author Rebuttal · Reviewer_Ccnu · 2026-04-02
> >
> > Thank you for the detailed response. My concerns are somewhat addressed. A few points I’ll mention:
> > - I appreciate the trajectory similarity metric. I think just for readers to build better intuition about TTC, the authors may benefit from a specific visualization on how latent trajectories relate and differ. For example, in Sudoku, you could learn a linear probe per timestep to map latent states to a set of ground-truth board properties that you track. If the training accuracy of the probes increase over timesteps, it may offer additional insight into why reasoning accuracy increases as planning horizon increases (Figure 5).
> > - Following up on the CoT response, since all models use CoT prompting, it would be interesting to conduct a 2x2 experiment (w and w/o CoT, w and w/o TTC) to isolate the effect of TTC. I understand that this may be expensive to run if the models were pretrained with CoT already, but only conducting this 2x2 at test-time may still offer some insights.

---

> > > ### Author Response · Authors · 2026-04-06
> > >
> > > Thanks to the reviewer for proposing this concrete experiment and for the insightful suggestion to analyze the representations of intermediate states.
> > >
> > > **1. Linear probe in sudoku.**
> > >
> > > We extract the intermediate latent states induced by the learned internal state transitions in TTC, and train a linear probe to decode them into per-cell digit predictions. We then measure the cell-level accuracy of these decoded states at different planning steps.
> > > |Step|Acc.|
> > > |---|---|
> > > |0|42.50|
> > > |2|42.96|
> > > |4|46.20|
> > > |8|50.38|
> > >
> > > We observe that decoding accuracy improves as the planning horizon increases, indicating that the latent states become progressively more aligned to the terminal states. We note that TTC layers are not directly supervised by the cell-level ground-truth (only linear probes are trained). The emergent increasing alignment between latent states and output-level groudtruth supports the reviewer's hypothesis and provides a mechanistic explanation for the observed gains in reasoning accuracy with longer horizons.
> > >
> > > We will include these results and corresponding visualizations in the final version to further strengthen the interpretability of TTC.
> > >
> > > **2. 2x2 ablation on CoT.**
> > >
> > > We reuse the fine-tuned checkpoints from our main experiments and apply a system prompt to disable CoT during inference. We manually verify that the adopted prompt can effectively suppress explicit CoT generation. The results on the AMC benchmark are shown below:
> > >
> > > |Models|w/ CoT|w/o CoT|
> > > |---|---|---|
> > > |w/o TTC|20.78|3.92|
> > > |w/ TTC|23.34|7.83|
> > >
> > > We observe that TTC consistently improves performance even without explicit CoT, nearly doubling accuracy compared to the baseline without TTC. This indicates that an internal planning brought by TTC can partially substitute for explicit reasoning traces to enhance reasoning capacity.
> > >
> > > Importantly, our goal is not to eliminate CoT. Explicit intermediate reasoning remains beneficial and often necessary for complex tasks. Rather, TTC is designed to **enhance the quality of each reasoning step** by incorporating lookahead planning and propagating long-horizon feedback into next-token predictions. As a result, TTC complements CoT by making reasoning more accurate and less reliant on lengthy or redundant test-time searching steps (e.g., ToT and RAP).

---

### Official Review · Reviewer_1xoj · 2026-03-13

**Soundness:** 2
**Presentation:** 2
**Significance:** 2
**Originality:** 2
**Overall Recommendation:** 2
**Confidence:** 3

**Summary:**

This paper introduces a Test-Time Control (TTC) layer designed to enhance reasoning capabilities in large language models by formulating reasoning as an optimal control problem. The proposed TTC-Net architecture integrates finite-horizon Linear-Quadratic Regulator planning into the model's forward pass, aiming to enable planning before prediction. The authors claim improvements in mathematical reasoning tasks, such as MATH-500 and AMC, through hardware-efficient implementations. The work positions itself as a novel approach beyond traditional test-time training methods.

**Compliance With Llm Reviewing Policy:**

Affirmed.

**Final Justification:**

The author has added many more experiments, and the original text needs to be significantly revised. As the original text is rather incomplete, I expressed my concerns.

**Key Questions For Authors:**

Refer to Weaknesses

**Limitations:**

The approach is evaluated only on constrained reasoning tasks, with no evidence of scalability to broader language understanding or generation challenges.
The computational overhead of the TTC layer is not thoroughly analyzed, leaving open questions about feasibility in resource-limited environments.
The paper does not address potential ethical or safety implications of deploying control-based reasoning systems in sensitive applications.
Ablation studies are insufficient to isolate the contribution of individual components, weakening the support for architectural choices.
The work assumes idealized conditions in simulations, with no discussion of robustness to noise or adversarial inputs in practical settings.

**Strengths And Weaknesses:**

Strengths：
The paper proposes an innovative integration of optimal control theory with neural network architectures, offering a fresh perspective on reasoning in language models.
Empirical results demonstrate performance gains on certain benchmarks, suggesting potential applicability in structured tasks.
The hardware-aware design attempts to address computational efficiency, which is a practical concern in large-scale deployments.

Weaknesses：
The methodological explanations are overly complex and lack clarity, making it difficult to assess the novelty and reproducibility of the approach.
Experimental evaluations rely heavily on synthetic or narrow datasets, raising doubts about generalizability to real-world scenarios.
The paper fails to provide sufficient comparisons with state-of-the-art baselines, undermining the validity of claimed advancements.
Technical details are buried in dense mathematical formulations, obscuring the core contributions and practical implementation steps.
The writing style is verbose and repetitive, reducing the overall readability and impact of the presentation.

---

> ### Author Rebuttal · Authors · 2026-03-31
>
> We thank Reviewer 1xoj for taking the time to review our work. However, we find that the comments are quite general and do not provide sufficient detail to identify specific concerns to address. We have nevertheless made our best effort to respond based on our interpretation of the feedback, as outlined below. We would greatly appreciate it if the reviewer could provide more specific and detailed comments to help us better address their concerns.
>
> **1. Writing complexity and clarity.**
>
> We welcome reviewer's constructive feedback on writing quality. However, we are not able to locate actionable issues from overly generic comments such as "methodological explanations are overly complex and lack clarity". These statements are not tied to any specific part of the paper.  We'd appreciate more specific locations and the exact sentences on possible confusion.
>
> ICML submissions are expected to be technical, precise, and formal. The mathematical formulation in our work is not ornamental, but essential for rigorously defining the control-based reasoning mechanism. Each component serves a specific purpose: Equations (1-3) establish the motivation, Equations (4-5) formalize forward dynamics, Theorem 3.2 ensures differentiability, and Theorem 3.3 derives hardware-efficient solvers. All results are accompanied by complete proofs. Hence we respectfully disagree on "Technical details are buried in dense mathematical formulations, obscuring the core contributions".
>
> We hope that technical rigor does not lead to lack of clarity. If specific sections or equations require clarification, we are happy to address them.  We will also add pseudocode and additional implementation details to further improve reproducibility.
>
> **2. Evaluation beyond reasoning datasets.**
>
> Our primary objective is to improve reasoning via architectural design. Mathematical datasets are widely recognized benchmarks for evaluating reasoning ability.
>
> To demonstrate generality beyond math reasoning, we evaluate our fine-tuned checkpoints on additional benchmarks:
>
> ||Llama + SFT|TTC-Net|
> |---|---|---|
> |Hellaswag|54.01|53.75|
> |Winogrande|74.74|72.09|
> |Arc challenge|51.79|59.22|
> |Mmlu|59.31|64.02|
> |Humaneval|32.32|49.39|
>
> Consistent with math results, TTC-Net shows strong effectiveness on reasoning-heavy tasks such as MMLU and HumanEval, while maintaining competitive performance on others.
>
> We further test our model upon Qwen family and the results are reported in the response to Reviewer AME7.
>
> **3. Comparisons with more baselines.**
>
> Our primary contribution lies in architecture-level improvements for LLMs. Accordingly, we focus on comparisons with mainstream architectural alternatives (e.g., Mamba, GDN, Samba), which represent competing sequence modeling approaches grounded in memory mechanisms.
>
> To broaden comparison, we include inference-time planning strategies such as Best-of-N (BoN) and Tree-of-Thought (ToT) [1], enabling direct comparison between internal planning (TTC-Net) and external search.
>
> |Method|Accuracy|Avg Token|
> |---|---|---|
> |BoN|22.43|20731.10|
> |ToT|21.08|22412.02|
> |TTC-Net (Ours)|23.34|2801.50|
>
> TTC consistently outperforms BoN and ToT in reasoning accuracy. More importantly, TTC produces substantially shorter outputs, indicating **significantly improved token efficiency and more direct reasoning trajectories**.
>
> [1] Yao et al., Tree of Thoughts: Deliberate Problem Solving with Large Language Models
>
> **4. Computational overhead.**
>
> We benchmark inference efficiency of TTC-Net. Due to hardware-efficient LQR solvers and kernel fusion, TTC introduces minimal computational overhead:
>
> |Model|Throughput (tokens/s)|
> |---|---|
> |Transformer|47.16|
> |TTC-Net ($T=8$)|45.77|
> |TTC-Net ($T=16$)|44.61|
> |TTC-Net ($T=64$)|39.43|
> |TTC-Net ($T=128$)|36.18|
>
> Throughput decreases moderately as horizon increases, while reasoning performance consistently improves, demonstrating a favorable efficiency-performance trade-off.
>
> **5. Additional ablation studies**
>
> We conduct ablations on: (1) time-homogeneous vs. time-heterogeneous parameterization, (2) TTC layer insertion interval, and (3) training horizon sampling strategy. Due to space limit, we leave the result table and analysis in the response to Reviewer ghUw. These ablation studies validate the key design choices described in our paper.
>
> **6. Limitations on robustness or safety.**
>
> Our work focuses on architectural improvements for reasoning, which is largely orthogonal to robustness and safety. That said, stronger reasoning may indirectly improve robustness by enabling models to better detect adversarial or malicious inputs.
>
> Robustness and safety are predominantly architecture-agnostic challenges. Existing methods can be directly applied to TTC-Net. Therefore, this concern does not undermine the validity or notable contribution of our work.

---

> > ### Author Rebuttal · Reviewer_1xoj · 2026-04-03
> >
> > Thank you for your Rebuttal. Thank you for solving most of my problems.  I think the experiments in the paper are rather weak. The author has added many more experiments, and the original text needs to be significantly revised. As the original text is rather incomplete, I expressed my concerns.

---

> > > ### Author Response · Authors · 2026-04-05
> > >
> > > We find it puzzling to state "the experiments are rather weak" after our rebuttal, especially without any concrete reference to missing evaluations or specific deficiencies.  As supported by other reviews, we believe that sufficient and targeted experiments that directly support the main claims have been included, instead of exhaustively covering unrelated directions. Our work focuses on architectural improvements for reasoning capability, and all empirical evaluations are aligned with this objective.  In our rebuttal, we have further included evaluations on another family of LLMs, additional benchmarks, a careful analysis of efficiency, and ablation studies on our core design choices.  Experiments orthogonal to our claims (e.g. "robustness to adversarial noise") is beyond the primary scope of this work. While interesting in their own right, including them could dilute the focus from properly validating our core contributions.
> > >
> > > We also respectfully disagree with the statement that the "original text needs to be significantly revised". Our core claims, technical results, and primary empirical findings remain valid and well-supported. The additional results provided in the rebuttal serve to further strengthen the paper, but do not indicate any fundamental issues requiring substantial revision.  We would appreciate to know what specific sections and/or paragraphs are unclear to this reviewer.
> > >
> > > From the initial review through the current exchange, we found the feedback remaining high-level, non-specific, and lacks actionable technical arguments (e.g., no pointers to specific sections, equations, or analyses).  We are still unable to identify what additional experiments are needed or which specific parts of the manuscript require revision based on the generic comments.  We would appreciate more constructive and concrete suggestions, as provided by other reviews, which can help us achieve more meaningful improvements in both evaluation and presentation.
> > >
> > > Overall, we believe that our response, together with the original submission, adequately addresses the concerns raised. We sincerely welcome clear and specific suggestions to further improve the paper.

---

### Decision · Program_Chairs · 2026-04-30

**Decision:**

Accept (regular)

**Comment:**

This paper formulates reasoning as an optimal control problem and introduces the Test-Time Control (TTC) layer, which performs finite-horizon LQR planning over latent states at inference time, supported by a hardware-efficient symplectic LQR solver implemented as a fused CUDA kernel. The paper received scores of 4, 4, 4 (Weak Accept) and 2 (Reject). Reviewer 1xoj's reject cited generic issues but did not reference specific section, equation, or experiment in the paper. The post-rebuttal response acknowledged most concerns were addressed. The paper presents a novel architectural idea with solid theoretical grounding, careful systems-level implementation, and empirical gains validated across multiple settings. I recommend weak accept.